# rStar-Coder: Scaling Competitive Code Reasoning with a Large-Scale Verified Dataset

**Yifei Liu**[1] **Li Lyna Zhang**[2][†] **Yi Zhu**[2]
**Bingcheng Dong**[2,3] **Xudong Zhou**[2,4] **Ning Shang**[2] **Fan Yang**[2] **Cheng Li**[1,◇] **Mao Yang**[2]

[1]University of Science and Technology of China  [2]Microsoft Research Asia
[3]Dalian University of Technology  [4]Shanghai Jiao Tong University

## Abstract

Advancing code reasoning in large language models (LLMs) is fundamentally limited by the scarcity of *high-difficulty* datasets, especially those with *verifiable input-output test cases* necessary for rigorous solution validation at scale. We introduce **rStar-Coder**, which significantly improves LLM code reasoning capabilities by constructing a large-scale, *verified* dataset of 418K competition-level code problems, 580K long-reasoning solutions along with rich test cases of varying difficulty. This is achieved through three core contributions: **(1)** we curate competitive programming code problems and solutions to synthesize new, solvable problems; **(2)** we introduce a reliable input-output test case synthesis pipeline that decouples the generation into a *three-step input generation* method and a *mutual verification mechanism for effective output labeling*; **(3)** we augment problems with high-quality, test-case-verified long-reasoning solutions. Extensive experiments on Qwen models (1.5B-14B) across various code reasoning benchmarks demonstrate the superiority of rStar-Coder dataset, achieving leading performance comparable to frontier reasoning LLMs with significantly smaller model sizes. On LiveCodeBench, rStar-Coder improves Qwen2.5-7B from 17.4% to an impressive 57.3%, and Qwen2.5-14B from 23.3% to 62.5%, surpassing o3-mini (low) by 3.1%. On the more challenging USA Computing Olympiad, our 7B model achieves an average pass@1 accuracy of 16.15%, outperforming the frontier-level QWQ-32B. rStar-Coder dataset is publicly available at https://huggingface.co/datasets/microsoft/rStar-Coder.

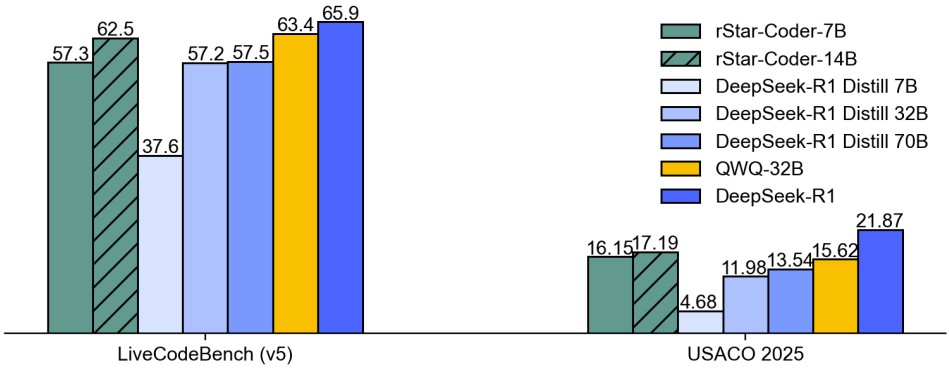

Figure 1: Pass@1 accuracy on code reasoning benchmarks.

---

[*]Yifei Liu, Bingcheng Dong, and Xudong Zhou completed this work during their internships at MSRA.
[†]Project leader: lzhani@microsoft.com
[◇]Corresponding author: chengli7@ustc.edu.cn

39th Conference on Neural Information Processing Systems (NeurIPS 2025).

# 1 Introduction

Recent large language models (LLMs) have made significant strides in reasoning, with models like OpenAI o1/o3 [17] and DeepSeek-R1 [9] showing strong performance on complex code problems. However, to further improve advanced code reasoning, a persistent challenge is the scarcity of large-scale, high-quality datasets that contain verifiable, *high-difficulty* programming problems requiring both *algorithmic thinking* and efficient code implementations.

Unlike datasets on math word problems, where solution correctness can be verified through simple rule-based matching against reference answers [9, 8], code reasoning requires executing solutions against diverse test cases to detect logical and implementation errors. These input-output test cases must differ in content, scale, and complexity while meeting problem-specific constraints. Critically, each test output must correctly label the expected result for its corresponding test input.

Obtaining such *reliable* and *verifiable* input-output test cases at scale remains highly challenging. Human-curated datasets, such as CodeContests [22] and TACO [20], provide high-quality problems but typically lack comprehensive test coverage, often including only a few simple test cases that fail to capture diverse input scales of edge conditions. Synthetic datasets such as WizardCoder [24], Magicoder [30], and KodCode [31] primarily target function-level code generation, where correctness can be verified with minimal or even no test cases. These datasets are therefore insufficient for advanced competitive-level code reasoning. While frontier LLMs may offer a scalable way for test case synthesis, two substantial challenges arise: generating semantically valid, constraint-aware inputs varying in scale and difficulty; and labeling correct outputs without ground truth solutions, particularly since synthetic problems inherently lack reference implementations.

In this paper, we introduce rStar-Coder, a novel approach that reliably constructs a large-scale, high-difficulty dataset for training advanced code reasoning LLMs. This dataset includes 418K unique competitive programming problems and 580K long-reasoning solutions, each verified through diverse synthetic test cases across difficulty levels. rStar-Coder incorporates three key components.

First, we curate and clean 37.7K expert-written problems with oracle solutions from competitive programming platforms (e.g., IOI, Codeforces) and use them as seeds to synthesize new, solvable problems. Unlike prior works [31, 24] that emphasize diversity, we prioritize solvability and correctness, since the seed set already spans a wide range of topics and difficulty levels. However, these problems are often too difficult for even frontier models like GPT-4o, and directly prompting the LLM with only the problem statement often results in invalid or unsolvable outputs. To address this, we design a structured prompt that incorporates both the problem statement and its oracle solution, guiding the model to understand core algorithmic concepts and generate novel yet solvable problems.

Second, we address the challenge of generating reliable and diverse input-output test cases for solution validation by decoupling the process into two stages. **(1)** First, we propose a three-step approach to generate valid test inputs of varying scale and complexity. We prompt GPT-4o to generate two utility functions for each problem: one for synthesizing semantically valid inputs with exposed scale-controlling parameters, and another for validating whether the inputs satisfy problem-specific constraints. We then sample input scale values ($10^0$ - $10^5$) for these scale-controlling parameters to cover a diverse range of complexities. Executing the two utility functions with these sampled scales yields diverse, constraint-satisfying test inputs. **(2)** Second, to reliably label outputs, we introduce a mutual verification mechanism. We sample multiple long-CoT solutions from a strong reasoning model (QWQ-32B), and accept both the test outputs and solutions if a majority produces consistent results across *all test inputs*. This mechanism is effective because incorrect solutions tend to diverge in errors, while correct solutions converge. Thus, consistent agreement among multiple solutions on identical outputs across diverse test inputs, particularly those of varying complexity, serves as strong evidence for the correctness of both the solution logic and the resulting outputs. Our ablation study demonstrate the effectiveness of this approach, achieving 96.8% accuracy in output labeling.

Finally, we augment curated expert-written problems with verified long-reasoning solutions. While these problems are of high-quality, their original solutions often lack detailed reasoning steps. Generating them directly with frontier LLMs is unreliable, as few simple original tests are insufficient for validation. To address this, we use our test generation method to produce diverse, constraint-aware inputs. Since curated problems include oracle solutions, we run them to get ground-truth outputs. We then prompt QWQ-32B for long-reasoning solutions, keeping only those that pass all generated tests.

Extensive experiments across different-sized LLMs (1.5B-14B) and diverse code reasoning benchmarks demonstrate the effectiveness of rStar-Coder, consistently improving the code reasoning capabilities of all base models to state-of-the-art levels, even with significantly smaller sizes. On LiveCodeBench, rStar-Coder improves the 14B model from 23.3% to 62.5%, surpassing R1-distill-70B and o3-mini (low) by +5.0% and 3.1%. And even the 1.5B model reaches 40.1%, outperforming both R1-distill-7B and GPT-4o. On the highly competitive USA Computing Olympiad, rStar-Coder-7B and rStar-Coder-14B outperform the frontier reasoning model QWQ-32B. Furthermore, rStar-Coder also generalizes remarkably well on standard code generation tasks like HumanEval and MBPP.

## 2 Related Works

**Instruction Code Data Synthesis**. Prior to the rise of reasoning-focused LLMs, code data synthesis methods [24, 30, 32] primarily targeted scaling instruction data to improve LLMs capabilities to generate code aligned with use intent. Efforts like Code Alpaca [4], WizardCoder [24], Magicoder [30] and WaveCoder [32] focused on synthesizing diverse and complex prompts by collecting seed code snippets and prompting LLMs for generating instructions and solutions. Recently, KodCode [31] scaled this paradigm by synthesizing 447K prompts from 12 sources using five distinct methods to ensure diversity and complexity. While effective for improving general instruction-following, these datasets show limited gains on reasoning-heavy code tasks [31, 16].

**Code Reasoning LLMs and Datasets**. Recent advances in code reasoning LLMs have garnered attention, with two main approaches: distilling long-reasoning solutions from frontier models like R1 and o3-mini [26, 2, 1], and applying reinforcement learning to high-difficulty code problems [25, 35]. Both face limitations. Distillation-based methods build datasets [26, 2, 28] by scaling solutions per problem but suffer diminishing returns due to limited diversity. RL methods struggle with unreliable evaluation, as comprehensive test cases are often lacking. To our knowledge, no prior work has systematically scaled algorithmic problem sets while providing robust, verifiable test case generation.

**Code Solution Verification**. Verifying the correctness of LLM-generated code solutions, especially for competition-level problems, remains highly challenging. A common approach is using generated test cases for validation. Prior work often uses the same LLM to produce both solutions and test cases, relying on self-consistency to select the most agreed-upon output [5, 11, 33], or comparing outputs to oracle solutions derived via exhaustive brute-force search [34]. However, the validity and diversity of test cases across difficulty levels have not been extensively explored, yet both are essential for robust evaluation. Our work addresses this challenge through a three-step input generation method and a mutual-verification mechanism that reliably labels both solutions and test cases.

## 3 Methodology

### 3.1 Collection of Competitive Code Problems

**Collection of expert-designed competitive coding problems**. We curate a seed dataset from publicly available resources, including programming competition websites and open datasets, where the problems and test cases are designed by human experts. This includes original problems, reference solutions, and available test cases from TACO [21], APPS [10], CodeContests [22], CodeContests-Python-Submission [14], CodeFroces from the OpenR1 project [26], and USA Computing Olympiad 2011-2023 (USACO) [27]. We also gather problems from the International Olympiad Informatics (IOI) spanning 2002-2023. As IOI problems are published in PDF format, we follow NuminaMath [19] and use Mathpix to convert them into LaTex. In total, we collect 57,215 problems (see Table 1). To ensure high quality, we remove duplicate problems across datasets and discard problems lacking reference solutions, resulting in 37,754 unique problems with at least one reference solution.

**Synthesis of new solvable and high-difficulty code problems**. Then, we use the 37,754 expert-written problems as seeds to synthesize new ones. These competitive programming problems span a wide range of algorithmic topics and data structures, but each type is sparsely represented. Therefore, unlike prior works that focus on maximizing the diversity between generated and seed problems [31, 24, 13], our focus is on generating problems that are both *solvable* and *high-difficulty*.

However, directly prompting LLMs (e.g., GPT-4o) with seed problems alone often fails to generate valid new problems, as even GPT-4o struggle to solve competition-level code problems. Without a deep understanding of the knowledge and reasoning skills being tested, the LLM lacks the necessary grounding to create problems that preserve both difficulty and solvability. To address this, we

Table 1: Summary of competitive-level programming problems. In total, we collect 418K verified problems, including 37.7K expert-designed and 380K synthetic problems.

| Source | Original Seed Questions | | Synthesized Questions | | Total |
|---|---|---|---|---|---|
| | # Original | # Verified | # Synthesized | # Verified | Verified |
| AIZU | 4302 | 2151 | 100712 | 17386 | 19537 |
| AtCoder | 2824 | 2080 | 88103 | 29096 | 31176 |
| CodeChef | 5232 | 3765 | 164646 | 46749 | 50514 |
| CodeWars | 4975 | 2515 | 108565 | 8520 | 11035 |
| GeeksForGeeks | 2680 | 2680 | 116809 | 37060 | 39740 |
| HackerEarth | 3657 | 1411 | 166385 | 48185 | 49596 |
| HackerRank | 860 | 854 | 35959 | 11707 | 12561 |
| LeetCode | 1516 | 754 | 32794 | 2282 | 3036 |
| CodeForces | 30049 | 20616 | 706289 | 170713 | 191329 |
| International Olympiad Informatics (IOI) | 626 | 444 | 24760 | 1767 | 2211 |
| USA Computing Olympiad (USACO) | 484 | 484 | 20610 | 7095 | 7579 |
| Total | 57,215 | **37,754** | 1,565,632 | **380,560** | **418,314** |

Figure 2: Examples of standard input-output test case pairs from competitive programming datasets. The first line indicates the scale of the test input (e.g., the size of 1D array), followed by each subsequent line representing the specific values for the input elements.

design structured prompts that include both the seed problem, its reference solution, and step-by-step synthesis instructions. The reference solution helps the LLM internalize the key algorithmic reasoning concepts involved in the seed problem. Specifically, we instruct the model to: (1) understand the seed problem and solution; (2) identify the reasoning and core knowledge being tested from the solution; (3) synthesize a new problem that test similar skills. Full prompt details are in Appendix Fig. 8.

In practice, incorporating reference solutions proves crucial for achieving high-quality synthesis. Even smaller models such as *Qwen2.5-72B-Instruct* perform well under this setup, though we adopt GPT-4o as the main generator to ensure higher reliability and success rates.

In totoal, we synthesize 1,565K new code problems as shown in Table 1. Compared to the original seeds, the synthesized problems still exhibit novelty by: (1) creating new contexts while maintaining the original algorithmic strategy (e.g., dynamic programming); (2) modifying or adding constraints to alter difficulty or computational complexity; (3) changing both context and constraints. These synthesized problems lack verifiable test cases and reference solutions, and some may be unsolvable. We address this through test case generation and mutual verification, as detailed in the next sections.

## 3.2 Test Case Generation

Checking whether a code solution runs without syntax or runtime errors is a straightforward way to verify correctness, but it's not sufficient. A correct solution must also be logically sound and bug-free. The standard approach is to use input-output test cases: given a test input (as shown in Fig. 2), a valid solution should produce the expected test output exactly. Effective test cases, therefore, should meet two key criteria. First, test inputs should cover a range of scales and complexities to evaluate both correctness and efficiency. For example, real-world code competitions often include large inputs (e.g., size $10^5$) to test whether a solution satisfies the algorithmic efficiency constraints and runs within time limits. Second, the test outputs must be accurate to support reliable validation.

However, obtaining such high-quality test cases is highly challenging. Curated datasets, though sourced from major coding platforms, typically provide only public test cases, which are often too simple to catch deeper logical errors. Synthetic problems, on the other hand, lack test cases entirely. Moreover, without ground-truth solutions, it becomes even harder to accurately label outputs, making reliable test case generation especially difficult. To address these challenges, we decouple the test case generation process into two stages: input generation and output labeling. Instead of prompting

**Test Input Constraints from Problem Statement:**
The first line contains single integer *n* (1<=≤<=*n*<=≤<=100000) — the size of the array.
The second line contains *n* integers *a*1,<=*a*2... *a*\*n* (1<=≤<=*a*\*i*<=≤<=10^9) — the elements of the array.

```python
# usage: print(generate_test_input(5))
# 2 3 5 6 4  ## possible generated test input
from cyaron import *
def generate_test_input(n):
    # Constraint check
    if not (2 <= n <= 100000):
        return None
    # Generate an array of size n with values between 1 and 1e9
    a = Vector.random(n, [(1, int(1e9))], mode=1)
    flat_a = sum(a, [])  # flatten to list of integers
    # Format input string
    input_lines = [str(n), ' '.join(map(str, flat_a))]
    return "\n".join(input_lines)
```

```python
# usage: print (validate_test_input("5\n2 3 4 5 6"))  # True
def validate_test_input(input_string):
    try:
        lines = input_string.strip().split('\n')
        if len(lines) != 2: return False
        n = int(lines[0])
        if not (2 <= n <= 100000):   return False
        a = list(map(int, lines[1].strip().split()))
        if len(a) != n:   return False
        if any(x < 1 or x > 10**9 for x in a):    return False
        return True
    except:
        return False
```

Figure 3: An example of LLM-generated utility functions for test input generation and validation.

an LLM to directly produce input-output pairs, we first introduce a three-step input generation method to create *valid and diverse* inputs of varying complexity that satisfy the problem's constraints. We then use a mutual verification mechanism to reliably label the corresponding test outputs.

### 3.2.1 Valid Test Input Generation with Varying Complexity

**Test case input format**. To generate valid test inputs across varying complexities, we first introduce the typical format of test inputs. As shown in Fig. 2, a test input in competitive programming is usually a multi-line string. The first line encodes one or more scale-controlling parameters (e.g., array size or grid dimension), and the following lines specify the actual input content. These strings are passed directly to the solution code, which must implement custom parsing logic to interpret them.

On top of that, generating valid inputs across different scales, however, is non-trivial due to two key challenges. First, although all inputs are string-encoded, their structure and semantics vary significantly across problems. For instance, a string might represent a 1D integer array, a list of strings, coordinates, or a 2D grid. Without understanding these structures, it is difficult to produce valid inputs. Moreover, problem statements often include constraints on valid input ranges, making naive prompting likely to produce invalid test inputs. Second, we aim to generate inputs of different complexities, yet naive prompting of LLMs tends to result in random and overly simple inputs ( see Fig. 5). To address these challenges, we propose a three-step approach as illustrated in Algorithm 1.

**Step 1: Generating utility functions for input generation and validation.** To produce high-quality test inputs that satisfy both the semantics and constraints of each problem, we prompt a frontier LLM (GPT-4o) to generate two utility functions per problem: one for test input generation and one for input validation (Fig. 3). This serves two purposes: **(1)** automatically producing well-structured inputs that satisfy problem constraints, and **(2)** exposing scale-controlling parameters (i.e., the first string line in Fig. 2) to enable flexible input sizing. Notably, direct LLM generation of input values often causes hallucinations. To reduce this, we allow GPT-4o to use CYaRon [◇], a reliable input data generation toolkit. As shown in Algorithm 1, given the problem description and CYaRon documentation, the LLM is asked to generate a GENERATE_TEST_INPUT function that uses scale parameters to call CYaRon for input construction, and a VALIDATE_TEST_INPUT function that parses the resulting input string and checks for constraint satisfaction. Prompt details are in Appendix Fig. 9.

**Step2: Defining input scale ranges**. From the scale-controlling parameters exposed by the GENERATE_TEST_INPUT function in Step 1, we define value scales for each parameter to control test case difficulty (e.g., $1$–$9\times10^0$ for easy cases, up to $10^5$ for hard cases). For example, in a 1D array, the scale denotes the number of elements (e.g., $10^5$ means 100K elements); in a 2D grid, it has two scale-controlling parameters for the number of rows and columns. We instantiate each parameter across its range and input the values into the generation function.

**Step3: Executing utility functions to produce valid test inputs**. Finally, for each instantiated scale-controlling parameters from Step 2, we invoke the GENERATE_TEST_INPUT function to generate a test input string. We then use the VALIDATE_TEST_INPUT function to verify whether each generated input string meets the constraints outlined in the corresponding problem statement. Only the inputs that pass validation are retained as valid test inputs.

---

[◇] https://github.com/luogu-dev/cyaron

---

**Algorithm 1** Three-Step Test Input Generation Algorithm

---

**Step 1: GPT-4o generates test input and validation functions with applying the CYaRon library.**

1: **function** GENERATE_TEST_INPUT($param_1, param_2, \ldots$)    $\triangleright$ $param_1, param_2, \ldots$ controls the input scales
2:     **return** None **If** parameters do not satisfy the problem constraints
3:     Generate input using the CYaRon library
4:     **return** input_string
5: **end function**

1: **function** VALIDATE_TEST_INPUT($input\_string$)
2:     Check whether input_string satisfies all constraints in the problem statements.
3:     **return** True if valid; otherwise False
4: **end function**

**Step 2: Define varies input scales**

1: for each $(param_1, param_2, \ldots, param_n)$ extracted from GENERATE_TEST_INPUT:
2: $\mathcal{C} = \{1, 2, \ldots, 9\} \cup \{10^i \mid 0 \le i \le e\}$

**Step 3: Execute functions to produce valid test inputs**

1: **for** each $(param_1, param_2, \ldots, param_n) \in \mathcal{C}$ **do**
2:     $input\_string \leftarrow$ GENERATE_TEST_INPUT($param_1, param_2, \ldots$)
3:     $valid \leftarrow$ VALIDATE_TEST_INPUT($input\_string$)
4:     Retain $input\_string$ **If** $valid$ is True
5: **end for**

---

**Augmenting difficult test inputs for seed problems**. In addition to generating test inputs for synthetic code problems, we also enhance the test inputs for original seed problems, which often feature either simple public test cases or none at all. By applying the above pipeline, we generate more diverse and challenging test inputs for these problems.

### 3.2.2 Mutual Verification for Test Output and Solution Labeling

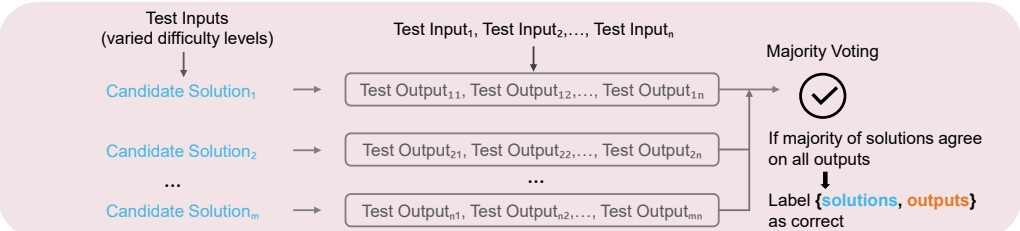

Figure 4: Mutual verification between candidate solutions and test outputs. Given a diverse set of inputs, if most solutions return the same outputs, both the solution and outputs are considered correct.

The next step is to reliably label each test input with the correct output. Since our synthetic test inputs come from two sources (i.e., the original seed problems and synthesized problems), we apply different strategies to achieve optimal labeling reliability for each.

**Labeling test output for seed problems**. For our augmented test inputs from seed problems, we simply execute the provided oracle solution on the input. Since the reference solution is assumed correct, its output serves as a the ground-truth label.

**Mutual verification for synthetic problems**. However, labeling test outputs for synthetic problems is quite challenging as there are no oracle solutions. To address this, we propose a simple yet effective mutual verification mechanism that identifies both correct test outputs and the solutions that produce them. As illustrated in Fig. 4, for each problem, we first sample 16 long-reasoning candidate solutions using a frontier reasoning model (QWQ-32B [29]). We then sample a diverse set of at least 50 test inputs with varying complexities. Each candidate solution is executed on this shared set of test inputs to generate the corresponding outputs. If a majority of these candidate solutions produce identical sets of outputs across this entire set of test inputs, then both these consistent sets of outputs and the candidate solutions that generated them are considered correct.

The effectiveness of mutual verification stems from the fact that incorrect solutions are more likely to diverge in errors than to converge on the exact same incorrect answer across multiple test inputs. If a majority of independently generated solutions produce identical results for a diverse set of inputs, it suggests they are not just randomly failing or misunderstanding the problem in different ways. Instead, their consistent agreement indicates successfully solved the problem, providing strong evidence for the correctness of both the solutions and their generated outputs.

### 3.3 Augmentation and Post-processing

**Seed problems augmentation**. Although our seed problems are expert-designed and high quality, their original oracle solutions often lack detailed reasoning steps. To address this, we rewrite the solutions to include rich reasoning patterns, such as self-reflection, which are essential for training advanced code reasoning LLMs. To verify correctness, we augment each seed problem with diverse test cases generated by our method, which are used for solution verification and filtering. Specifically, we use QWQ-32B to generate 16 long CoT solutions per problem and retain only those that pass all tests. For particularly challenging problems where QWQ-32B fails to produce a correct solution, we follow prior work [26, 2] and retain all generated solutions to include more diverse and potentially correct intermediate reasoning steps in the training data.

**Post-Processing for synthetic data**. We also clean the synthetic data to ensure high quality. First, we remove unsolvable or overly difficult problems, which may result from hallucinated generation or limitations of the frontier reasoning model. The mutual verification mechanism naturally acts as an effective filter. Specifically, if fewer than 60% of solutions agree on the outputs, the problem is discarded. For problems synthesized based on Codeforces problems, we use the original `cf_rating` to identify hard problems (`cf_rating` > 1600) and adjust the threshold to 40% to include harder problems. This process effectively filters out unreliable problems, as shown in Table 1. After filtering, we retain 380K verified synthetic problems. For these 380K problems, we initially have 2.25M long CoT solutions, which are too large for efficient fine-tuning. We reduce this by executing all solutions and retaining only the fastest one per problem based on CPU execution time.

**Decontamination**. To ensure fair and unbiased evaluation, we follow prior works [15] and perform data decontamination by removing any problems that overlap (16-gram) with evaluation benchmarks: HumanEval, HumanEval+, MBPP, MBPP+, LiveCodeBench, and USACO 2025. We adopt a 16-gram threshold because shorter n-grams (e.g., 8- or 10-gram) often produce false positives in competitive programming problems, which naturally share common phrases in I/O descriptions and problem statements (e.g., "the sum of n overall all test cases"). After decontamination, our dataset contains 418K problems with extensive test cases, totaling 580K question–solution pairs.

## 4 Experiments

### 4.1 Setup

**Training setup**. Using our 580K dataset, we fine-tune Qwen2.5-Coder instruct models [16] at 1.5B, 7B, and 14B sizes for 6 epochs using the AdamW optimizer, a batch size of 96, and a max sequence length of 16k. The learning rate is 4e-5 with cosine decay. Training is accelerated with FlashAttention-2 [7] and DeepSpeed ZeRO-0. Specifically, the 1.5B and 7B models are trained on 8 MI300X AMD GPUs, while the 14B model uses 32 MI300X GPUs.

**Evaluation benchmarks**. We evaluate on diverse benchmarks, including LiveCodeBench [18] v5, which features new problems from LeetCode, AtCoder, and Codeforces (2024.8-2025.2), and a challenging new benchmark from the USA Computing Olympiad 2025 (**USACO**), containing 12 Olympiad problems across bronze to platinum tiers. These problems test a broad spectrum of algorithmic and commonsense reasoning. In addition, we test generalization abilities by evaluating on popular code generation benchmarks: HumanEval [6], HumanEval+ [23], MBPP [3], and MBPP+ [23].

**Baselines and Inference Settings**. We compare against two strong baselines: (1) leading frontier LLMs, including GPT-4o, Claude 3.5 Sonnet, DeepSeek-R1, and QWQ-32B; and (2) state-of-the-art code reasoning models fine-tuned on large-scale long-CoT code datasets. Specifically, in addition to DeepSeek-R1 distilled models, we compare with Bespoke-Stratos (fine-tuned on 17k traces) [28], OpenThinker (114k) [28], OlympicCoder (100k) [26], Nvidia OCR-Owen (736k) [2], and OpenThinker2 (1M) [28]. Notably, OCR and OpenThinker2 are trained with significantly larger datasets than ours. For inference, we set temperature to 0.6, maximum output length to 32k tokens.

Table 2: Results of rStar-Coder and frontier reasoning LLMs on diverse benchmarks. We outperform on both reasoning-heavy and general code generation tasks with significantly smaller model sizes. *Gray text in ( ) indicates our absolute score improvements over the base model.*

| Model | LiveCodeBench | HumanEval | HumanEval+ | MBPP | MBPP+ |
|---|---|---|---|---|---|
| GPT-4o | 30.0 | 92.7 | 87.2 | 87.6 | 72.2 |
| Claude3.5-Sonnet | 32.0 | 92.1 | 86 | 91 | 74.6 |
| OpenAI o3-mini-2025-01-31 (low) | 59.4 | - | - | - | - |
| QWQ-32B | 63.4 | 95.6 | 89.8 | 92.2 | 76.5 |
| OpenAI o1 | 63.4 | - | - | - | - |
| DeepSeek-R1 | 65.9 | 96.3 | 90.9 | 95.3 | 81.2 |
| Gemini-2.5-Pro | 69.2 | - | - | - | - |
| 1.5B Long-CoT coder reasoning LLMs | | | | | |
| DeepSeek-R1-Distill-Qwen-1.5B | 16.9 | 66.3 | 61.8 | 57.1 | 48.5 |
| **rStar-Coder-1.5B** | **40.1** (+33.6) | **88.4** (+17.7) | **81.7** (+16.5) | **74.1** (+8.9) | **60.8** (+1.5) |
| Bespoke-Stratos-7B(Bespoke-Stratos-17k) | 16.2 | 75.1 | 69.1 | 73.6 | 59.8 |
| OpenThinker-7B (OpenThoughts-114k) | 25.5 | 80.5 | 75.7 | 80.9 | 68.2 |
| OpenThinker2-7B (OpenThoughts2-1M) | 37.4 | 92.7 | 87.8 | 86.9 | 73.9 |
| DeepSeek-R1-Distill-Qwen-7B | 37.6 | 89.6 | 83.7 | 78.4 | 66.7 |
| DeepSeek-R1-Distill-LLaMA3-8B | 39.6 | 85.9 | 79.7 | 62.7 | 52.8 |
| OlympicCoder-7B (OpenR1-codeforces-100k) | 40.9 | 82.1 | 76.9 | 80.0 | 66.4 |
| OCR-Qwen-7B-Instruct (OCR-736k) | 51.3 | - | - | - | - |
| **rStar-Coder-7B** | **57.3** (+39.9) | **95.9** (+7.5) | **90.8** (+6.7) | **87.9** (+4.4) | **74.0** (+2.3) |
| 14B-70B Long-CoT reasoning LLMs | | | | | |
| DeepSeek-R1-Distill-Qwen-14B | 53.1 | 96.1 | 89.8 | 87.4 | 74.1 |
| OCR-Qwen-14B-Instruct (OCR-736k) | 59.4 | - | - | - | - |
| Bespoke-Stratos-32B | 48.9 | 95.5 | 89.8 | 93.8 | 77.5 |
| DeepSeek-R1-Distill-Qwen-32B | 57.2 | 95.9 | 89.9 | 92.8 | 78.7 |
| OpenThinker-32B (OpenThoughts-114k) | 54.1 | 94.8 | 89.2 | 94.1 | 78.4 |
| OlympicCoder-32B (OpenR1-codeforces-100k) | 57.4 | 90.0 | 85.2 | 86.7 | 71.3 |
| OCR-Qwen-32B-Instruct (OCR-736k) | 61.7 | - | - | - | - |
| DeepSeek-R1-Distill-LLaMA-70B | 57.5 | 96.5 | 90.7 | 91.9 | 77.1 |
| **rStar-Coder-14B** | **62.5** (+39.2) | **95.9** (+6.3) | **89.6** (+2.4) | **91.4** (+5.2) | **77.3** (+4.5) |

To mitigate performance variance inherent in single-run evaluations, we sample 16 solutions per problem and report the average pass@1 accuracy for all benchmarks.

## 4.2 Main Results

**Results on diverse code benchmarks**. Table 2 presents the performance of rStar-Coder compared to state-of-the-art reasoning models. We highlight three key observations: **(1)** By scaling with our high-quality dataset, rStar-Coder significantly improves LLMs code reasoning capabilities, achieving performance comparable to frontier reasoning LLMs with substantially smaller model size (1.5B-14B). For instance, Qwen2.5-Coder-1.5B, originally at only 6.5% accuracy on LiveCodeBench, improved dramatically to 40.1% with rStar-Coder, even outperforming GPT-4o and R1-Distill-Qwen-1.5B. Moreover, larger models see even greater gains: rStar-Coder-7B achieves 57.3%, surpassing R1-Distilled-Qwen-32B. rStar-Coder-14B reaches 62.5%, outperforming all open-source baselines, including OCR-32B and R1-Distill-70B, and even surpass o3-mini by 3.1%. **(2)** Dataset quality matters more than size. Though OCR [2] and OpenThinker-2 [28] use much larger datasets (736K and 1M vs. our 580K), rStar-Coder performs significantly better–by +6% and 19.9% at 7B, and even surpasses their 32B models at 14B. **(3)** Despite not being tailored for general code generation, rStar-Coder generalizes remarkably well. rStar-Coder consistently improves the performance on HumanEval, HumanEval+, MBPP, and MBPP+ of all base models to state-of-the-art levels. Notably, rStar-Coder-7B achieves performance on par with Claude3.5-Sonnet, showing that strong reasoning data can generalize effectively beyond its original domain.

**Results on challenging Olympiad programming**. We further evaluate on USACO 2025, which features highly challenging algorithmic problems across four tiers: Bronze to Platinum. Unlike standard competitive questions, these problems require grounded and often creative reasoning, especially at the Gold and Platinum levels, where deep algorithmic insight and creativity are essential. As shown in Table 3, the benchmark is extremely difficult. Even OpenAI o3 scores only 32.03% and fail on all Platinum problems. Despite the difficulty, our 7B and 14B models perform competitively, outperforming the QWQ-32B. Notably, QWQ-32B generated the long-reasoning solutions in our dataset. That even rStar-Coder-7B surpasses it highlights the strength of our data: diverse, competitive problems and verified high-quality reasoning enable smaller models to rival far larger ones.

Table 3: rStar-Coder performs competitively on USACO 2025 with much smaller model sizes.

| Model | Avg. | Bronze | Silver | Gold | Platinum |
|---|---|---|---|---|---|
| OpenAI-o3 | 32.03 | 72.91 | 28.12 | 27.08 | 0 |
| DeepSeek-R1 | 21.87 | 58.33 | 22.91 | 6.24 | 0 |
| QWQ-32B | 15.62 | 43.75 | 12.5 | 6.25 | 0 |
| 7B-8B Long-CoT coder reasoning LLMs | | | | | |
| OpenThinker-7B | 0 | 0 | 0 | 0 | 0 |
| OlympicCoder-7B | 0.52 | 2.08 | 0 | 0 | 0 |
| OpenThinker2-7B | 4.16 | 16.67 | 0 | 0 | 0 |
| DeepSeek-R1-Distill-Qwen-7B | 4.68 | 18.75 | 0 | 0 | 0 |
| **rStar-Coder-7B** | **16.15** | **47.92** | **4.17** | **12.5** | **0** |
| 14B-70B Long-CoT coder reasoning LLMs | | | | | |
| DeepSeek-R1-Distill-Qwen-14B | 8.85 | 33.33 | 2.08 | 0 | 0 |
| OpenThinker-32B | 9.37 | 35.42 | 2.08 | 0 | 0 |
| OlympicCoder-32B | 9.89 | 35.42 | 0 | 4.17 | 0 |
| OpenThinker2-32B | 14.06 | 39.58 | 12.5 | 4.17 | 0 |
| DeepSeek-R1-Distill-Qwen-32B | 11.98 | 39.58 | 4.17 | 4.17 | 0 |
| DeepSeek-R1-Distill-LLaMA-70B | 13.54 | 43.75 | 8.33 | 2.08 | 0 |
| **rStar-Coder-14B** | **17.19** | **47.92** | **12.50** | **8.33** | **0** |

Table 4: Ablation on curated-only vs. synthetic-only subsets proves the value of each source.

| Model | LiveCodeBench | HumanEval | HumanEval+ | MBPP | MBPP+ |
|---|---|---|---|---|---|
| DeepSeek-R1-Distill-7B | 37.6 | 89.6 | 83.7 | 78.4 | 66.7 |
| *Synthetic-only subset 7B* | 46.8 | 89.6 | 83.6 | 86.5 | 72.7 |
| *Seed-only subset 7B* | 49.7 | 93.7 | 88.2 | **91.0** | 73.7 |
| **rStar-Coder-7B** | **57.3** | **95.9** | **90.8** | 87.9 | **74.0** |

## 4.3 Ablation Study and Analysis

**Quality of seed and synthetic data sources**. To evaluate the contribution of each data source, we conduct SFT on 7B scale using only expert-written seed problems and only synthetic problems with their corresponding solutions. As shown in Table 4, our 7B model finetuned on either the curated or synthetic subset significantly outperforms the R1-Distill-7B model on both reasoning-intensive and general code generation benchmarks, showing both sources are highly effective. While each subset individually underperforms compared to our full dataset, the results indicate that curated and synthetic data provide complementary benefits, and their combination yields the strongest performance.

**Effectiveness of mutual verification**. Mutual verification enables reliable labeling of test outputs without oracle solutions. To evaluate its effectiveness, we randomly sample 64 expert-written seed problems with oracle solutions and collect all their test inputs (**3,150** in total). For each input, we label test outputs using mutual verification: long CoT solutions from QWQ-32B are executed to produce outputs, and majority voting is applied to determine the final label. These outputs are then compared against ground-truth outputs obtained by running the oracle solutions.

As a baseline, we prompt GPT-4o to directly generate input-output pairs, following prior work [31, 12] (see Prompt in Appendix Fig. 10). As shown in Table 5 (Left), mutual verification achieves 96.8% accuracy, while the GPT-4o baseline yields only 12.7%, highlighting the reliability from our method. To further validate scalability, we extend the evaluation to 1,024 problems (**27,613** test cases in total), where mutual verification still maintains 92.8% accuracy, confirming its robustness at larger scales.

Table 5: Ablation study on the reliability of synthetic test cases. *Left*: Our mutual verification ensures the high accuracy test output labeling without oracle solutions. *Right*: Compared to directly generating test inputs with LLMs, our three-step approach significantly improves input quality.

| Method | Accuracy |
|---|---|
| GPT-4o verification | 12.7% |
| **Mutual verification** | **96.8%** |

| Method | LiveCodeBench | | | |
|---|---|---|---|---|
| | Average | Easy | Medium | Hard |
| GPT-4o prompting | 42.9 | 87.3 | 54.1 | 10.6 |
| **Three-step input generation** | **44.6** | **87.7** | **56.5** | **12.6** |

**Effectiveness of test input generation**. An key component in rStar-Coder is the three-step test input generation method, which produces constraint-satisfying and diverse inputs critical for mutual

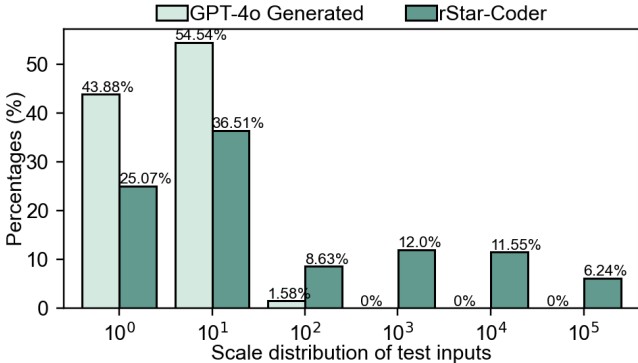

Figure 5: rStar-Coder generates more diverse and larger-scale test inputs, while directly LLM prompting yields smaller and simpler cases.

verification and accurate labeling. To evaluate its impact, we sample 150K synthetic problems and generate inputs using (1) our method and (2) a GPT-4o prompting baseline (Appendix Fig. 11). We then apply the same mutual verification process and fine-tune Qwen2.5-Coder-7B-Instruct on both resulting datasets for 3 epochs. As shown in Table 5 (Right), our method yields much higher results on LiveCodeBench across all difficulty levels, demonstrating the importance of diverse and complexity-aware inputs for stronger verification, which we further evaluate in the next experiment.

**Ablation on the test input scales**. To verify the correctness of code solutions, test inputs need to span a wide range of difficulty. Our three-step generation method explicitly controls input scales to achieve this. To evaluate the effectiveness, we sample 1K problems from our dataset and plot the distribution of test input scales. For comparison, we also generate inputs using direct GPT-4o prompting (Appendix Fig. 11). As shown in Fig. 5, our method produces inputs that evenly cover the range from easy ($10^0$) to very hard ($10^5$), while GPT-4o generated inputs are concentrated in the easier range ($10^0$–$10^2$), with no input scales exceeding $10^3$. This demonstrates the superiority of our method in generating more challenging and diverse test cases.

## 5 Conclusion

In this work, we present rStar-Coder to construct a large-scale, high-quality dataset for advancing LLMs in competitive code reasoning. By introducing a reliable test case generation method, we address the core challenge in generating verified solutions for high-difficulty code problems. Built upon expert-curated seeds, our approach enables large-scale synthesis and augmentation, resulting in 418K verified competitive-level problems and 580K long-reasoning solutions. Extensive experiments across different-sized LLMs (1.5B-14B) and diverse code reasoning benchmarks demonstrate the superiority of rStar-Coder, achieving performance comparable to QWQ-32B while consistently outperforming existing code reasoning models at the same model scale. As future work, we plan to further expand the dataset by curating more problems and scaling up synthesis and verification.

## Acknowledgement

This work was supported by grant from the National Key R&D Program of China (No. 2024YFB4505701).

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

# A    Appendix

## A.1    Additional Results and Discussions

Table 6: Ablation on scaling dimensions for the 7B model shows that expanding problem diversity is more effective and efficient than only increasing the number of solutions per problem.

| Dataset | Unique Problems | Data size | LiveCodeBench | USACO 2025 |
|---|---|---|---|---|
| Seed Problems× 1 solutions | 37.7K | 37.7K | 40.8 | 0.52 |
| Seed Problems× 8 solutions | 37.7K | 302K | 51.1 | 3.64 |
| Seed Problems× 16 solutions | 37.7K | 603K | 54.7 | 10.41 |
| **rStar-Coder dataset** | 480K | 580K | **57.3** | **16.15** |

**Analysis of different scaling dimensions.** Most existing code reasoning datasets focus on scaling the number of long CoT solutions per problem based on a limited set of code problems [2, 26, 28]. In contrast, rStar-Coder emphasizes not only scaling expert-curated solutions but also expanding the number of unique code problems. To compare the effectiveness of these two scaling dimensions, we conduct a controlled experiment using 37.7K expert-designed problems. We vary the number of long-CoT solutions per problem (1, 8, and 16), while keeping the problem set fixed. The 16-solution setting yields 603K examples, already surpassing the total size of rStar-Coder-580K.

As shown in Table 6, both scaling solution count and problem diversity improve reasoning performance. However, scaling only the number of solutions yields diminishing returns and becomes less efficient. For example, our 580K dataset, with broader problem coverage, achieves significantly better results on reasoning-heavy benchmarks like LiveCodeBench and USACO than the 603K dataset derived from scaling solutions alone. Notably, for training efficiency, rStar-Coder-580K includes only one solution per synthesized problem, and we plan to scale this further in future work.

Table 7: Ablation on scaling up synthetic data size.

| Dataset | LiveCodeBench | HumanEval | HumanEval+ | MBPP | MBPP+ |
|---|---|---|---|---|---|
| Seed+50K synthetic | 54.0 | 94.4 | 88.3 | **91.2** | **76.5** |
| Seed+100K synthetic | 55.1 | 94.2 | 87.9 | 90.6 | 75.4 |
| Seed+200K synthetic | 56.2 | 93.8 | 88.0 | 89.2 | 74.5 |
| Seed+all synthetic (rStar-Coder dataset) | **57.3** | **95.9** | **90.8** | 87.9 | 74.0 |

**Scaling the size of synthetic data**. We further analyze the impact of scaling the amount of synthetic data. Specifically, we incrementally add synthesized problems to a fixed set of seed problems while keeping other factors constant. As shown in Table 7, reasoning performance on LiveCodeBench improves consistently with larger synthetic datasets, demonstrating the effectiveness and scalability of our data generation pipeline.

Table 8: Core skills evaluated at each tier of USACO, from https://usaco.guide/.

| Difficulty | Core Skills Evaluated |
|---|---|
| Bronze | simulation, complete search, sorting, greedy |
| Silver | binary search, comparators, graphs, trees, floodfill, prefix sums, bitwise operators |
| Gold | dynamic programming, disjoint set union, spanning trees, Euler tour, combinatorics |
| Platinum | segment tree, range queries, binary jumping, sweep line, convex hull, flows |

**USACO 2025 Benchmark**. The USA Computing Olympiad (USACO) is a prestigious algorithmic programming competition for high school students in the United States, consisting of four difficulty levels: Bronze, Silver, Gold, and Platinum. Each level contains a set of challenging problems that test algorithmic thinking and implementation skills, making USACO a valuable benchmark for evaluating the reasoning and problem-solving capabilities of large language models. We adopt the USACO 2025 problem set as a benchmark, which includes 12 problems, with three from each level. Following [27], we evaluate zero-shot pass@1 for each difficulty level and report the average pass@1 across all four levels as the final score.

**Discussions and Limitations**. Our approach relies on substantial GPT resources for synthesizing code problems and test inputs. Many generated problems are discarded after mutual verification due to being invalid or unsolvable. Additionally, we have observed that some competition problem

statements do not explicitly provide constraints but instead imply them through context. Since our current method primarily depends on frontier LLMs to interpret the problem statements, it is not yet capable of handling such cases. Furthermore, our pipeline is currently specialized for competitive programming problems, and adapting it to more open-ended software engineering tasks (e.g., bug fixing, pull request generation) would likely require additional design and modifications. We leave these challenges to future work.

**Broader Impact**. rStar-Coder enables the development of stronger code reasoning models by providing large-scale, high-difficulty problems with verifiable test cases. This supports progress in algorithmic reasoning, and AI-assisted programming. However, similar to the other reasoning LLMs, our rStar-Coder could also generate misleading, harmful or hallucinated outputs. We recommend careful consideration of potential misuse during training and deployment, and encourage future work on improving the reliability and safety of code reasoning systems.

## A.2    Code Problem Types

According to the input-output format, competitive programming problems can be divided to two types: standard input-output based problems and function-based problems.

### A.2.1    Standard Input-Output Based Problems

Standard Input-Output based problems require the solution code to read from standard input, parse the actual input content, and write results to standard output. As shown in the example problem 6, the first line indicates the number of test cases, and each of the subsequent lines contains two integers representing a single test case.

Code 1 provides an example of evaluation logic for this type of problem, utilizing Python's subprocess module to execute the solution code under time and memory constraints. Since the input is read from standard input, it can be treated as a plain string.

To generate input samples of varying scales for these problems, we employ GPT-4o and DeepSeek-V3 to synthesize two utility functions—generate_test_input and validate_test_input—using the prompt shown in 11. The generate_test_input function takes the specified problem scale as input and produces a formatted input string, which is subsequently checked for validity by validate_test_input.

Figure 6: Standard input-ouput based problem example

## Standard Input/Output Problem Example

You have two positive integers $a$ and $b$.
You can perform two kinds of operations:

- $a = \lfloor a/b \rfloor$
- $b = b + 1$

Find the minimum number of operations required to make $a = 0$.

**Input Format**

The first line contains a single integer $t$ ($1 \le t \le 100$) — the number of test cases.
Each of the following $t$ lines contains two integers $a$ and $b$ ($1 \le a, b \le 10^9$).

**Output Format**

For each test case, print a single integer: the minimum number of operations required to make
$a = 0$.

**Example Input**

```
6
9 2
1337 1
1 1
50000000 4
991026972 997
1234 5678
```

**Example Output**

```
4
9
2
12
3
1
```

**Solution Code:**

```python
t = int(input())
test_cases=[tuple(map(int, input().split())) for _ in range(t)]
def min_operations(a, b):
        if b == 1:
            b += 1
            operations = 1
        else:
            operations = 0
            min_ops = float('inf')
            for increment in range(100):
                current_a = a
                current_b = b + increment
                current_operations = operations + increment
                while current_a > 0:
                    current_a //= current_b
                    current_operations += 1
                    min_ops = min(min_ops, current_operations)
            return min_ops

for a, b in test_cases:
    print(min_operations(a, b))
```

Figure 7: Function-based problem example

---

**Function-based Problem Example**

Given a number $s$ (in string form), find the smallest number (without leading zeros) that can
be obtained by rearranging its digits.
**Example 1:** Input: s = "846903" Output: 304689
**Example 2:** Input: s = "55010" Output: 10055
**Starter Code:**

```
class Solution:
        def minimum_Number(self, s):
        # Code here
```

---

Listing 1: Evaluation for Standard Input/Output Problems

```python
def run_solution(solution_code, input_data):
  try:
    def set_cpu_affinity():
      resource.setrlimit(resource.RLIMIT_AS, (MAX_MEMORY, MAX_MEMORY))
    process = subprocess.Popen(
        ['python3', '-c', solution_code],
        stdin=subprocess.PIPE,
        stdout=subprocess.PIPE,
        stderr=subprocess.PIPE,
        text=True,
        preexec_fn=set_cpu_affinity,
        )
    stdout,stderr=process.communicate(input=input_data,timeout=TIMEOUT)
    success = process.returncode == 0
    return stdout if success else stderr , success
  except:
    return "Exception", False
  finally:
    if process and process.poll() is None:
        process.kill()
```

### A.2.2 Function-Based Problems with Starter Code

### A.2.3 Function-Based Problems

Function-based problems provide a starter function or class definition as part of the problem statement.
The solution is written by completing the specified function, which is then invoked with deserialized
inputs. As shown in the example problem 7, each test input is represented as a parameter dictionary,
and the expected output is compared against the return value of the function.

Code 2 presents an example of evaluation logic for this type of problem. During evaluation, the input
string—stored in a serialized format—is first deserialized into structured data (e.g., integers, arrays, or
strings), which is then passed as arguments to the solution function through dynamic code execution.
The return value is captured and compared to the expected output to determine correctness.

To generate inputs of varying scales for these problems, we employ GPT-4o and DeepSeek-V3 to
synthesize two utility functions. The generate_test_input function produces a list of arguments
in serialized format, which is then deserialized and validated by validate_test_input to ensure
correctness and consistency.

Listing 2: Evaluation for Function-Based Problems

```
from pyext import RuntimeModule
method_name = in_outs["fn_name"]
inputs = [json.loads(line) for line in inputs.split("\n")]

mod = RuntimeModule.from_string("tmp_sol", "", sol)
obj = mod if "class Solution" not in test else mod.Solution()
method = getattr(obj, method_name)
output = method(*inputs)
```

# B Prompts

Figure 8: New code problem synthesis prompt.

---

**New Code Problem Synthesis Prompt in rStar-Coder**

I will provide you with a programming problem along with its solution. Your task is to create a new, transformed programming problem based on the original one.
You need to complete the following steps:
1. Analyze and understand the original problem and its solution. Identify the reasoning steps (e.g., Step 1, Step 2, Step 3) and summarize the knowledge points tested in the original problem.
2. Design a new problem that is similar to the original one and can be solved using the same knowledge points. If you reference any conditions or descriptions from the original problem, rewrite them clearly and avoid phrases like "as in the original problem".

- Provide two example test cases to demonstrate the new problem.
- Ensure that the complexity of the new problem is well-designed by specifying appropriate input constraints.

Your output should follow this format:
## Part 1: Original Problem and Solution Analysis
Step 1: [Describe the first step of reasoning]
Step 2: [Describe the second step of reasoning]
...
Knowledge Points: [Summarize the knowledge points tested, separated by commas if there are multiple]

## Part 2: New Problem Problem Description: [Describe the new problem clearly in natural language. Ensure it doesn't directly copy from the original problem description. Avoid phrases like "as in the original problem".]

Input Format: [Specify the input format]

Output Format: [Specify the output format]

## Part 3: Example Test Cases
Input: [Input for test case 1]
Output: [Expected output for test case 1]

Input: [Input for test case 2]
Output: [Expected output for test case 2]

Given Problem: {question}
Given Solution: {solution}

---

Figure 9: Test Input Generation Prompt for Standard Input/Output based Problems

**rStar-Coder: Test Input Generation Prompt for Standard Input/Output based Problems**

I will provide you with a programming problem description, and your task is to **generate standardized test input samples using the CYaRon library**.
You need to complete the following steps:
**1. Parse the constraints** on the input from the problem description, such as the range of input data, specific input constraints, etc.
**2. Write a function `generate_test_input`** using the CYaRon library to randomly generate test inputs based on a specified problem size. The function should validate that the parameters fall within the specified constraints. If any parameter is out of range, the function should return None. If the parameters are valid, generate a random test input and return an input string (`input_string`).
**3. Write a function `validate_test_input`** to verify whether the generated test input satisfies the requirements specified in the problem description. This includes checking the input data type and constraints parsed in step 1, such as range and other conditions. The function should take `input_string` as input and return a boolean (`True/False`).
**Part 1: Parse Input Constraints**
Specify the input constraints as described in the problem.
**Part 2: Code for Test Input Generation**

```
import cyaron as cy
def generate_test_input(<param1>, <param2>, ...):
    # Check if parameters meet constraints
    if not (<condition1>) or not (<condition2>):
        return None
    # Generate input using CYaRon
    input_data = [
        ...
    ]
    return "\n".join(map(str, input_data))
```

**Part 3: Code to Validate Test Input**

```
def validate_test_input(input_string):
    # Validation logic
    return <boolean>
```

**Given Problem:** {question}

---

Figure 10: Ablation study: Prompt for directly generating test input-output pairs with GPT-4o

**Ablation study: Prompt for directly generating test input-output pairs with GPT-4o**

I will provide you with a programming problem description, and your task is to generate test inputs and outputs for the problem.
You need to generate **50 test inputs and outputs pair** that effectively **verify the correctness of the core logic** and **assess the time complexity** of the solution. Ensure that your test cases cover a **diverse range of problem scales**, including **edge cases, small inputs, and large inputs** that push the problem's constraints.
Your output should follow this JSON format:

```
{
  "test_inputs": [
    {
      "idx": 0,
      "input_string": <test input 0>,
      "output_string": <test output 0>
    },
    ...
  ],
}
```

**Given Problem:** {question}

Figure 11: Ablation study: Prompt for directly generating test input with GPT-4o

**Ablation study: Prompt for directly generating test input with GPT-4o**

I will provide you with a programming problem description, and your task is to generate test inputs for the problem.

You need to generate **50 test inputs** that effectively **verify the correctness of the core logic** and **assess the time complexity** of the solution. Ensure that your test cases cover a **diverse range of problem scales**, including **edge cases, small inputs, and large inputs** that push the problem's constraints.

Your output should follow this JSON format:

```
{
  "test_inputs": [
    {
        "idx": 0,
        "input_string": <simple test input 0>,
      },
      ...
      ],
}
```

**Given Problem:** {question}

