# OpenReview forum: "rStar-Coder: Scaling Competitive Code Reasoning  with a  Large-Scale Verified  Dataset"
_NeurIPS.cc/2025/Conference — NeurIPS 2025 poster_

### Official Review · Reviewer_H9E1 · 2025-06-15

**Clarity:** 3
**Significance:** 3
**Originality:** 2
**Rating:** 4
**Confidence:** 4

**Summary:**

The paper addresses the scarcity of high-difficulty, verified code reasoning datasets for LLMs. It constructs 418K competitive problems and 580K long-reasoning solutions with a three-step approach: synthesizing problems, generating diverse test cases via GPT-4o, mutual verification, and augmenting solutions. Experiments show Qwen models fine-tuned on rStar-Coder outperform larger baselines on LiveCodeBench, USACO, and general code tasks, proving data quality drives reasoning gains.

**Questions:**

1. Is it reasonable to use 16-gram deduplication? Do you have real examples of overlapping 15-grams?
2. The core contribution of the paper seems to be a combination of WizardCoder and CodeT, with a larger volume. Could you explain the difference between the proposed method and WizardCoder / CodeT?

**Ethical Concerns:**

["NO or VERY MINOR ethics concerns only"]

**Final Justification:**

My only concern is that 16-gram deduplication may introduce unwanted contamination. Although the author frames this as a trade-off, I believe the deduplication of the test set should be enforced with the utmost strictness rather than treated as a trade-off.

Overall, I recognize that this paper makes a valuable contribution to the community, and I am therefore maintaining my weak-positive score.

**Limitations:**

This article mainly discusses how to construct a dataset of long-term CoT in code, which seems to be more suitable for the NeurIPS D&B Track.

**Quality:**

3

**Strengths And Weaknesses:**

#### Strengths
1. The paper proposes a large-scale, high-quality dataset with 418K coding problems and rigorous validation.
2. The paper proposes innovative test case generation using GPT-4o and cross-validation for reliability.
3. The paper demonstrates that smaller models can outperform larger ones, proving that data quality over quantity matters.

#### Weaknesses
1. As a paper with a dataset as its main contribution, the authors only uploaded some sample data.
2. The method relies heavily on GPT resources for problem synthesis, increasing costs and dependency.
3. Mutual verification will filter out the most difficult questions because the model can not generate the correct solution or test case.

---

> ### Author Rebuttal · Authors · 2025-07-31
>
> We sincerely appreciate your thoughtful review and your recognition of our dataset construction, test case design, and model performance results. Below we address the specific questions and comments:
>
> > As a paper with a dataset as its main contribution, the authors only uploaded some sample data.
>
> **Response**: Thank you for your question. Due to the Neurips anonymity policy, we are unable to share access to the full dataset during the rebuttal phase.  However, the complete dataset has already been carefully prepared. It has been used internally and externally in ongoing experiments, demonstrating its quality and usability in practice. In addition, we will include a detailed taxonomy in the appendix to provide a clearer global view of our dataset. Below is a preview,  organized along three key dimensions: algorithmic task types, file size breakdown, and solution token length distribution.
>
> |Algorithmic Task Types|
> | :--:   |
> | 2-sat, binary search, bitmasks, brute force, chinese remainder theorem, combinatorics, constructive algorithms, data structures, dfs and similar, divide and conquer, dp, dsu, expression parsing, fft, flows, games, geometry, graph matchings, graphs, greedy, hashing, implementation, interactive, math, matrices, meet-in-the-middle, number theory, probabilities, schedules, shortest paths, sortings, string suffix structures, strings, ternary search, trees, two pointers |
>
>
> |Total file size (compressed)| seed problems&solutions | seed test cases| synthetic problems&solutions|synthetic test cases|
> | :--:   |:--:   |:--:   |:--:   |:--:   |
> |**442.1G** | 12G|167G |6.1G | 257G|
>
> |Token length distribution (Solution Traces)| # of solutions  | percentage|
> | :--:   |:--:   |:--:   |
> |<4k|118593|20.3%|
> |4k-8k|111421|19.1%|
> |8k-16k|233002|39.9%|
> |>16k|120748|20.7%|
>
>
>  We are fully committed to releasing the complete dataset to advance open-source code reasoning research and benefit the broader community.
>
> >The method relies heavily on GPT resources for problem synthesis, increasing costs and dependency.
>
> **Response**: Thank you for the comment. As acknowledged in Appendix A (page 13, discussions and limitations section), our method currently relies on GPT resources for problem and test case synthesis. However, we believe this is a cost-effective and scable alternative to fully human-curated pipelines, which are often prohibitively expensive and difficult to scale. Looking ahead, we are optimistic about incorporating smaller open-source models as their reasoning capabilities continue to improve.
>
> >Mutual verification will filter out the most difficult questions because the model can not generate the correct solution or test case.
>
> **Response**: Thank you for the insightful comment. Indeed, mutual verification may filter out some of the most difficult problems when the model fails to generate consistent correct solutions. This reflects a current trade-off between dataset difficulty and correctness: for problems with low consistency ratios, it is challenging to reliably determine whether the issue stems from an ill-posed or unsolvable problem, or simply from high difficulty. To ensure data quality, we conservatively filter out such cases.
>
> We view this as an opportunity for future improvement. As reasoning model capabilities continue to advance, mutual verification can become more tolerant or be complemented by stronger coding agents. These improvements will help retain harder problems with greater confidence and further advance high-difficulty code reasoning.
>
> > Is it reasonable to use 16-gram deduplication? Do you have real examples of overlapping 15-grams?
>
> **Response**: Thank you for the thoughtful question. We take data contamination and train-test leakage very seriously and made careful design choices to minimize it.
>
> While many existing works (s1 for math [1], qwen2.5-coder for general code tasks [2]) use smaller thresholds like 8-gram or 10-gram for deduplication, we found that such settings lead to a high false positive rate in competitive programming problems. Many problems naturally share common patterns, especially in I/O description or problem statement. For example, common false positive cases like **"input the input is given in the following"**, **"the sum of n overall all test cases"**, **"be considered correct if its absolute or relative"**, **"your task is to find the number of"**.  As a result, short n-gram match often do not indicate true duplication.  We will include these concrete examples in the final version to illustrate this issue.
>
> As a result, we empirically tuned several n-gram thresholds and found that 16-gram is a good balance. Based on your suggestion, we also tested 15-gram filtering, which slightly increased the number of flagged overlaps from 312 to 464 out of 580K samples.  This suggests that our current deduplication is already conservative, and the overall duplication rate remains negligible (<0.1%).
>
> |benchmark|removed samples under 16-gram| removed samples under 15-gram|
> | :--:   |:--:   |:--:   |
> |Livecodebench | 216| 336|
> |Humaneval+| 45| 49|
> |MBPP+| 4| 4|
> |USACO 2025| 47| 75|
> |total| 312| 464|
>
> We appreciate the suggestion and will include a more detailed analysis of n-gram deduplication strategies in the final version.
>
> > The core contribution of the paper seems to be a combination of WizardCoder and CodeT, with a larger volume. Could you explain the difference between the proposed method and WizardCoder / CodeT?
>
> **Response**: Thank you for your question. While our approach also uses LLMs for dataset synthesis, it differs fundamentally from WizardCoder and CodeT in three key aspects: **task domain and difficulty**, **synthesis methodology**, and **quality control mechanisms**.
>
> WizardCoder focuses on general-purpose coding prompts, where each prompt can typically be solved with simple function-level code. In contrast, we target **competition-level programming problems**, a much more challenging domain that requires significant algorithmic reasoning and diverse test case coverage.
> These problems are significantly harder to generate, and cannot be created via unconstrainted prompt mutation alone.  Therefore, we use a solution-conditioned generation approach: given a seed problem and its step-by-step solution, the model produces new but solvable problems of comparable difficulty. This ensures both validity and challenge, which is essential for advancing reasoning skills.
>
> CodeT directly generates test input-output pairs using LLMs. While efficient, this assumes the LLM can fully solve the problem to produce both valid inputs and correct outputs, which is unrealistic for complex algorithmic tasks and prone to hallucinations.
>
>
> Our method **explicitly decouples** input and output generation. For test inputs, we use a **three-step generation** process: the model first generates *generate_test_input* and *validate_test_input* functions, then generates inputs by varying scale parameters, this yields inputs with diverse difficulty and guaranteed validity. For test outputs, we **do not rely on the LLM to generate them directly**. Instead, we apply mutual verification: a stronger reasoning model produces multiple candidate solutions, and outputs are retained only if they achieve agreement through execution-based majority voting.
>
> These mechanisms ensure the resulting dataset is not only larger but qualitatively superior: solvable, challenging, and robustly validated.
>
> We hope this clarifies that our work is not merely a scaled-up combination of WizardCoder and CodeT, but a new pipeline designed specifically for high-difficulty code reasoning, with clear methodological and empircal advances.
>
> > This article mainly discusses how to construct a dataset of long-term CoT in code, which seems to be more suitable for the NeurIPS D&B Track.
>
>  **Response**: Thank you for the suggestion. While our work introduces a high-quality dataset, we believe the core contribution lies in the **novel methodology** for synthesizing and verifying complex code reasoning problems at scale. Specifically, we address two fundamental challenges in competitive code reasoning:
>
> (1) how to synthesize new **solvable problems** that remain challenging and non-trivial, and
>
> (2) how to construct **diverse and reliable test cases** that cover varying input complexities while ensuring correctness.
>
> To tackle these, we introduce a solution-conditioned problem synthesis method, a three-step test input generation pipeline, and a mutual verification mechanism based on majority voting across model-generated solutions. These innovations enable the construction of high-quality problems and test cases that go well beyond the capabilities of prior approaches such as WizardCoder and CodeT.
>
> Furthermore, our focus is on **code reasoning**, a high-level cognitive capability that remains  **underexplored yet important** for advancing LLM intelligence. This makes rStar-Coder a valuable resources for LLM open-research community.
>
> Our empirical results show that training on this dataset **substantially improves LLM code reasoning ability** across multiple strong baseline models with much smaller model sizes. This highlights not only the utility of the dataset, but also the broader significance of our methodology for advancing LLM reasoning capabilities.
>
> We believe this integration of **methodological innovation**, **core research relevance**, and **impact on reasoning performance** makes our work well aligned with the goals of the main NeurIPS track.

---

> > ### Comment · Reviewer_H9E1 · 2025-08-06
> >
> > Thanks for your reply. Most of my concerns have been addressed. However, based on my experience, 16-gram may miss a lot of contamination samples, especially for test sets such as humaneval and mbpp that are a few years old, so I kept my score unchanged.

---

> > > ### Author Response · Authors · 2025-08-06
> > > **Response to Reviewer H9E1**
> > >
> > > Thank you for your valuable feedback! We’re glad to hear that most of your concerns have been addressed.
> > >
> > > Regarding the n-gram setting, we agree that for datasets focusing on general function-level code tasks, using a smaller n (i.e., a lower threshold) may be more appropriate. However, for competitive coding problems with lengthy problem descriptions (as illustrated in our response to other reviewers), using smaller n-grams such as 8-gram tends to produce many false positives. These false positives often arise from common code patterns rather than true contamination.
> > >
> > > To better illustrate this, we provide below some keywords detected by 8-gram on HumanEval and MBPP. As shown, many correspond to frequent, generic descriptions in code tasks and are unrelated to the core problem context.
> > >
> > > |# of cases detected| key phrases detected based on HumanEval|
> > > | :--:   |:--:   |
> > > |148| your task is to count the number of|
> > > |35| your task is to determine if it is|
> > > |35|task is to determine if it is possible|
> > > |28|are given a string s. your task is|
> > > |28|given a string s. your task is to|
> > > |24|is a natural number greater than 1 that|
> > > |17|is to determine if it is possible to|
> > >
> > > |# of cases detected| key phrases detected based on MBPP|
> > > | :--:   |:--:   |
> > > |11| the number of times it occurs in the|
> > > |8| to find the difference between the sum of|
> > > |4|write a function that counts the number of|
> > > |3|to find the length of the longest palindromic|
> > > |3|substrings with the sum of digits equal to|
> > >
> > > Based on these observations, we tuned the n thresholds and believe that using a 16-gram strikes a better balance between effectively detecting contamination and minimizing false positives.
> > >
> > > Thank you again for your insightful comments, which have helped us clarify this important aspect.

---

### Official Review · Reviewer_6vKP · 2025-07-02

**Clarity:** 1
**Significance:** 3
**Originality:** 2
**Rating:** 2
**Confidence:** 4

**Summary:**

The paper presents rStar-Coder, a dataset for tuning LLMs for competitive code reasoning. The paper claims a new input-output test generation method to augment existing competitive programming problems and a generated set of competitive programming codes with augmented inputs. The authors use the augmented code set to fine-tune several Qwen-based LLMs that performs better than other existing LLMs tuned for code synthesis tasks.

**Questions:**

$\ $

**Ethical Concerns:**

["NO or VERY MINOR ethics concerns only"]

**Limitations:**

Please see the comment in the weaknesses section.

**Quality:**

2

**Strengths And Weaknesses:**

Strengths:

* The problem that the paper tackles is interesting; as competitive problems are difficult for human programmers, they represent a good benchmark for LLMs. Generating correct input/output pairs that can be used in downstream LLM tasks, if done right, can lead to improved reliability of such codes.


* The approach uses LLMs to generate two functions for testing the algorithms: input generator and output checker. This construction is more likely to yield structurally valid inputs than directly prompting LLMs to do it. Further this approach can help human developer better understand and inspect the quality of the generated instances.

* The experimental results indicate that the approach can perform better than other LLMs tuned for coding tasks. Some solutions give more solved "Gold" level solutions for USACO'25 than even frontier models. What are the problems that were successfully solved and what insights did you obtain from studying the solutions? Discussing these problems and solutions in greater detail in the paper would be helpful for the readers to understand the full extent of the technique.


Weaknesses:

* Incorrect use of terminology. The standard definition of verification/verified pertaining to program code in software engineering and formal methods entails that the program will output the correct result for _all_ inputs [1,2]. The paper uses the terms verified and verification to mean that the programs pass _some_ inputs, i.e., a set of generated tests, which is likely to cause confusion. Ensuring the program correctness on all inputs is a significantly more difficult problem than the one tackled here. Similarly, the term mutual verification is similarly misleading. As it corresponds to majority voting between multiple tests, it should be simply called that way. Further, Section 3.2 mixes the terms 'verify' and 'validate', which are technically separate. Getting clear definitions and aligning with the standard ones is necessary for effectively communicating the research and putting the contributions of the paper in the proper context.

* The assumption that the solution of majority of generated samples is the correct solution is not fully confirmed by the paper and is still distant from the statement the paper advertises that the solutions in your dataset are verified. While the corresponding ablation study in the paper is a good starting point, and 95+% agreement is reasonably high probability for those sampled seed programs. However, confirming this hypothesis should involve both more sample seed programs (including reporting per-seed variances) and a qualitative study of generated solutions. A point of concern is that while checking independently obtained solutions could work when the sources of implementation are independent, with LLMs we cannot relay on that property, due to common training code bases so hidden dependencies cannot be excluded.

* The paper's limitations section should include the detailed discussion of which tasks the proposed approach/dataset is good for and for which it is not. Currently 'Limitations' are represented as a short paragraph in the appendix, but should be clearly marked in the main paper.

* Missing related work. One of the key contributions of this work is an approach to generate tests for augmenting LLM code generation. However, the paper does not cite any of the many recent techniques for test generation in software engineering. I suggest that the authors look into paper published at recent ICSE, FSE, ISSTA conferences and/or recent surveys e.g., [3,4]. Some other related techniques that use LLMs to output complex-input generators are explored in https://arxiv.org/pdf/2501.19282 .

* Writing can be improved. Proof-read the paper to catch common spelling errors (e.g., "totoal") and stylistic errors (e.g., "this" without following clause).



[1] https://en.wikipedia.org/wiki/Formal_verification

[2] Search-based Program Synthesis. 2018. https://dl.acm.org/doi/10.1145/3208071

[3] Software testing with large language models: Survey, landscape, and vision, 2024.

[4] Large language models for software engineering: A systematic literature review, 2024.

---

> ### Author Rebuttal · Authors · 2025-07-31
>
> **General Response to Reviewer 6vKP and Clarifications**:
>
> We sincerely thank the reviewer for the detailed and thoughtful feedback. Your comments, especially from a software engineering and formal methods perspective, helped us clarify our contributions and improve the paper's precision and rigor.
>
> **1. Clarifications on the use of verification terminology**: We acknowledge the reviewer’s concern regarding our use of the term “verification” and agree on the importance of aligning with established terminology in SE and formal methods, where “verification” often implies exhaustive, formal correctness across all inputs, which is a rigorous standard that we do not claim to meet.
>
>
> In our work, "verification" refers to an empirical process that mimics how human beings validate correctness via execution and consensus, rather than formal proof. This usage follows common practice in the LLM literature [1,2,3], where "verification" often denotes execution-based validation mechanisms that offer strong empirical reliability. Our intent is not to suggest formal guarantees, but rather to describe a practical and effective validation framework for long-CoT solutions and their corresponding test cases for competitive programming tasks. In the context of LLM code reasoning, this empirical rigor already marks a significant advance over prior work[4,5]. To avoid confusion, we will explicitly clarify this distinction in the revision.
>
>
> [1] DeepCoder: A fully open-source 14B coder at o3-mini level, https://www.together.ai/blog/deepcoder
>
>
> [2] OpenAI. Introducing SWE-Bench verified we're releasing a human-validated subset of swe-bench that more reliably evaluates AI models' ability to solve real-world software issues, 2024d. https://openai.com/index/introducing-swe-bench-verified/
>
> [3] DeepMath-103K: A large-scale, challenging, decontaminated, and verifiable mathematical dataset for advancing reasoning
>
> [4] ACL 2025, Best Paper Aware at DataWorld @ICML 2025, KodCode: A Diverse, Challenging, and Verifiable Synthetic Dataset for Coding
>
> [5] Huggingface, https://huggingface.co/blog/open-r1/update-3
>
>
> **2. Core contributions of our work**: We would like to reiterate the core contributions of our work: we present a **scalable, high-quality dataset generation pipeline** for constructing **high-difficulty competition-level** code reasoning datasets, a resource that is currently **highly scarce** and difficult to scale through human curation.
>
> Unlike general code generation tasks, competitive programming problems are fundamentally more challenging. They require multi-step algorithmic reasoning, careful planning, and precise validation against diverse edge cases. Yet, most existing datasets in this domain remain limited in both scale and quality, particularly due to the lack of **reliable and diverse test cases** that are essential for constructing high-quality training dataset.
>
> To address this, we introduce a pipeline that: (i) synthesizes new and solvable competition-level problems; (ii) generates parameter-controlled test inputs with varying difficulty; (iii) reliably label the test outputs and the long-cot solutions through mutual agreenment among multiple reasoning traces.
>
> Our goal is not just scaling data, but about **reliably scaling hard data**. This has been positively recognized by the other reviewers, such as **"The authors present a robust, parameter-controlled input generation toolkit that produces diverse and constraint-valid test cases, going beyond previous approaches that merely rely on LLMs to hallucinate inputs"**, **"rStar-Coder contributes a large-scale, publicly released corpus, which is likely to become a valuable resource for both training and evaluating code-reasoning LLMs"**, **"The paper proposes innovative test case generation using GPT-4o and cross-validation for reliability"**,
> **"The paper demonstrates that smaller models can outperform larger ones, proving that data quality over quantity matters"**.
>
> **3. Clarification of possible misunderstandings**: We would also like to clarify possible misunderstanding reflected in the review.
>
> 1) Code reasoning ≠ code synthesis. Our work targets code reasoning for high-difficulty competitive problems—not generic synthesis. Unlike conventional coding problems, code reasoning is particularly important for competitive coding problem. As shown in this paper, general code synthesis /LLM tuned for coding tasks is much less effective in this code domain, which is of high difficulty. Framing our contribution as "tuning for synthesis" misses the core challenge and motivation.
>
> 2) Not general test case augmentation. We also wish to correct the impression that our focus is not general test case augmentation, but a more specific, thus also more effective,  test case synthesis targeting code reasoning  validation. This differs from augmenting general downstram tasks with hallucinated inputs.
>
> 3) More than a benchmark. The reviewer refers to competitive problems as “a good benchmark,” which slightly misrepresents our contribution. Our goal is not just to benchmark existing models, but to enable training and evaluation on hard, verifiable, and diverse tasks through a generative solution pipeline.
>
>
>
> We hope our clarifications have provided a clearer explanation of the key insights and contributions of our approach. Given the novelty and demonstrated effectiveness of this approach, we believe in the value of our work. We welcome any further questions or suggestions you may have. Thank you again for your time and consideration.
>
> Below, we address your specific comments.
>
>
> > The assumption that the solution of majority of generated samples is the correct solution is not fully confirmed by the paper and is still distant from the statement the paper advertises that the solutions in your dataset are verified...
>
> **Response**: We appreciate your insightful comment. To further validate the effectiveness of our mutual verification strategy,  we conduct an additional experiment. We sample 1024 seed problems and use our mutual verification to label the test outputs for 27613 test inputs. We then compare the labeled test output against ground-truth test output to compute the correctness ratio. As shown in the table below, we observe an even higher correctness ratio of 97.1% across these samples, further demonstrating the reliability of our approach.
>
>
> |exp| #seed problems|  #test inputs|correctness of test outputs|
> | :--:   |:--:   |:--:   |:--:   |
> |Original ablation | 64| 3150| 96.8%|
> |**Additional ablation**|**1024** |**27613** |**97.1%**|
>
> Regarding the concern about the independence of LLM-generated solutions: while outputs may share dependencies due to overlapping training data, we believe that incorrect solutions tend to diverge in their errors rather than consistently converge. This makes it unlikely for multiple erroneous traces to produce identical outputs across diverse test cases. Consequently, majority agreement remains a reliable empirical signal of correctness, even if not formally independent.
>
> We believe this additional evidence further justifies mutual verification as a practical and effective mechanism for quality control, while recognizing that it provides empirical rather than formal guarantees.
>
> >The paper's limitations section should include the detailed discussion of which tasks the proposed approach/dataset is good for and for which it is not. Currently 'Limitations' are represented as a short paragraph in the appendix, but should be clearly marked in the main paper.
>
> **Response**: Thank you for the suggestion. We will move the discussion of limitations from the appendix into the main paper and expand it as recommended. Specifically, we will clarify that our proposed method and dataset are primarily designed for competitive programming problems, with the goal of advancing LLM reasoning capabilities. They are not intended for general software engineering tasks such as debugging, refactoring, or maintenance of real-world codebases. We appreciate the reviewer’s suggestion to make this distinction clearer.
>
> >Missing related work. the paper does not cite any of the many recent techniques for test generation in software engineering. I suggest that the authors look into paper published at recent ICSE, FSE, ISSTA conferences and/or recent surveys e.g., [3,4]. Some other related techniques that use LLMs to output complex-input generators are explored in https://arxiv.org/pdf/2501.19282 .
>
> **Response**: Thank you for the suggestion. We appreciate the reviewer’s point about better connecting our work to the broader software engineering (SE) literature on test generation. In our original submission, we focused primarily on LLM-driven data augmentation and acknowledge the omission of relevant SE research on automated test generation.
>
> We will revise the related work section to include representative papers from ICSE, FSE, ISSTA, and the suggested work [https://arxiv.org/pdf/2501.19282], which explores LLMs for input generation.
>
> We also wish to clarify a key distinction: our goal is not general-purpose test generation, but scalable, parameter-controlled test input synthesis tightly coupled with long-chain-of-thought solutions for competitive programming tasks. Unlike traditional SE settings, we do not assume access to complete program specs or ground-truth oracles. Our contribution lies in adapting test generation to support reliable long-cot supervision in high-difficulty code reasoning, where standard SE assumptions often break down.
>
> We thank the reviewer again and will incorporate these additions into the final version.
>
> Finally, we hope these responses address your concerns and clarify any confusion, and we will incorporate them in the revisions. Thank you again for your valuable feedback and suggestions, and we kindly ask you to consider re-evaluating our work.

---

> > ### Comment · Reviewer_6vKP · 2025-08-08
> >
> > Thank you for your detailed response. Several comments below:
> >
> > > Clarifications on the use of verification terminology
> >
> > While I understand that some works in the machine learning community has recently started to use the term verification to mean "passing the test case", the original point still stands. Such terminology can (and likely will, in the future) lead to confusion and should not be reinforced (as currently is from the title to abstract to main body of the paper).
> >
> > > Core contribution
> > > Unlike general code generation tasks, competitive programming problems are fundamentally more challenging.
> >
> > I can agree with your argument only to some extent. While competitive programming tasks poses the challenges you describe, focusing on those tasks significantly narrow down the search space of possible programs compared to general program generation (e.g., with respect to I/O, data structures, and error handling) offsetting the overall problem difficulty.
> >
> > The part of the response discussing "the clarification of possible misunderstanding" shows rather some differences in expectation from the problem statement:
> >
> > 1) (LLM-based) reasoning about code vs code synthesis: in the form of problem statement, code reasoning is an instance of code synthesis problem (in the form of algorithms, the approaches for traditional code synthesis differ from LLM-based approach).
> >
> > 2) the point that the test case synthesis is more effective in part because the problem is narrowed down to the specific kind of problems for code reasoning; while it has some merit and empirically are helpful for evaluated cases, the extent these tests exercise the code under test is unclear and the insight of how extensive they are is missing.
> >
> > 3) thanks for pointing out this point. To bring us to the same page, the "benchmark" used in that part of the review was meant to refer to a good application for exercising today's or future LLMs (not just evaluate existing ones)
> >
> > > an additional experiment. We sample 1024 seed problems and use our mutual verification to label the test outputs for 27613 test inputs
> >
> > Thanks for sharing the results. It would be interesting to understand the insight behind the observation "we believe that incorrect solutions tend to diverge in their errors rather than consistently converge".
> >
> > > Limitations and Related work
> >
> > Thank you for the response. Let me point out in light of "Unlike traditional SE settings, we do not assume access to complete program specs or ground-truth oracles" that in traditional SE setting both complete specs and ground-truth oracles are typically missing as well (often known as the "oracle problem", see e.g. [1] for common issues); in comparison, complete specs/oracles are more prevalent in formal verification literature, but often challenging to scale to complicated software -- another part of the reason why the proper terminology matters here.
> >
> > [1] The Oracle Problem in Software Testing: A Survey. IEEE TSE May 2015.

---

> ### Author Response · Authors · 2025-08-05
> **Kindly inviting further discussion**
>
> Dear Reviewer 6vKP，
>
> As the reviewer-author discussion phase is approaching its end, we would like to kindly check whether you have had a chance to read our response, particularly the parts regarding the clarification of the term “verification” and the possible misunderstandings.
>
> We sincerely appreciate the time and effort you have already dedicated to reviewing our paper. If there are any points that remain unclear or if you have further questions, we would be very grateful for the opportunity to clarify.
>
> Thank you again for your thoughtful review.

---

> ### Author Response · Authors · 2025-08-08
> **Response to Reviewer 6vKP**
>
> Thank you for your follow-up and continued engagement despite the limited time in the discussion phase.
>
> We would like to respectfully clarify that our work focuses on **high-difficulty competitive programming problems, which require complex algorithmic reasoning and multi-step solution validation with test input-output pairs at varying difficulty levels**. This focus fundamentally differs from traditional software engineering tasks such as general test input generation or program synthesis for software maintenance, leading to different challenges and evaluation criteria.
>
> Regarding your request for deeper insights into our mutual verification results, we observe empirically that incorrect solutions tend to diverge in their errors rather than converge, making majority voting a reliable heuristic for correctness. While this remains an empirical approach rather than a formal guarantee, in our previous response, our large-scale experiment involving over 27,000 test inputs showed a 97.1% correctness rate, supporting its practical effectiveness.
>
> We appreciate your careful reading and thoughtful feedback. Much of your comments focus on terminology and conceptual framing, and we believe our experiments and clarifications adequately addressed the core methodological concerns. We remain open to further suggestions and discussions.
>
> Thank you again for your time.

---

### Official Review · Reviewer_gnwi · 2025-07-02

**Clarity:** 3
**Significance:** 3
**Originality:** 2
**Rating:** 4
**Confidence:** 5

**Summary:**

The authors present rStar-Coder, a new pipeline and dataset aimed at pushing code-generation and code-reasoning abilities of LLMs. They start from 37.7 K expert-written competitive-programming problems and, through structured prompting that includes oracle solutions, synthesize an additional 380 K solvable tasks—yielding 418 K problems with 580 K long chain-of-thought solutions in total.

A key innovation is a two-stage test-case workflow: (i) a three-step input generator that exposes scale-controlling parameters and validates constraints, and (ii) a mutual-verification scheme where multiple solutions must agree on every output—achieving 96.8 % labeling accuracy.

Fine-tuning models on rStar-Coder dramatically boosts performance. For example, the Qwen2.5 7B model improves from 17% to 57.3% pass@1 on LiveCodeBench.

**Questions:**

1. Beyond the manually curated seed problems, your pipeline still relies on transforming existing tasks to create new ones. Could you clarify why this dependence is necessary? Is the method infeasible for problems that lack any reference solution, and does this imply that fully unconstrained scaling-up of code datasets is out of reach?
2. Could you explain how your approach could be extended to other domains—­for instance, to SWE-bench bug-fixing tasks or front-end/UI coding challenges—­where the problem format and test-case design differ markedly from competitive-programming questions?
3. Did you perform deduplication between the training corpus and each evaluation set to prevent train–test leakage? If so, please describe the procedure and its impact on the reported results.
4. How much of the observed performance gain stems from the presence of chain-of-thought supervision versus the increase in data number? Have you evaluated models trained on the same problems without CoT traces, and if so, what were the outcomes? If CoT is the dominant factor, does generating additional problems without reasoning traces yield any benefit at all?
5. The `validate_test_input` function itself could contain errors. What safeguards or verification steps do you employ to ensure that this component is correct?

**Ethical Concerns:**

["NO or VERY MINOR ethics concerns only"]

**Final Justification:**

The rebuttal addresses most of my questions. My key concern remains that verifying test correctness with multiple candidate solutions has been covered by prior work, which limits the incremental contribution of this submission. I am keeping my score unchanged.

**Limitations:**

The authors did not include a Limitations section in the paper. My suggestion is to refer to the Questions part above.

**Quality:**

3

**Strengths And Weaknesses:**

Strengths:
1. The authors present a robust, parameter-controlled input generation toolkit that produces diverse and constraint-valid test cases, going beyond previous approaches that merely rely on LLMs to hallucinate inputs. This allows candidate solutions to be evaluated over a much broader range of scenarios. Their ablation study further validates the effectiveness of this wide-ranging test coverage.
2. rStar-Coder contributes a large-scale, publicly released corpus—many of its solutions include chain-of-thought reasoning—which is likely to become a valuable resource for both training and evaluating code-reasoning LLMs.

Weaknesses:
1. Beyond the 37.7 K manually-curated seed problems, additional examples are produced by transforming or perturbing those same seeds. Consequently, scalability still hinges on acquiring fresh, high-quality seeds; without them, the method cannot generate truly novel problem families.
2. rStar-Coder’s problems are overwhelmingly competitive-programming style (algorithmic puzzles with full I/O control). It is unclear how well the pipeline generalises to more heterogeneous software-engineering tasks—e.g. SWE-bench bug-fixing, class-level code synthesis, or front-end/web-UI challenges—where test-case design and solution formats differ substantially.
3. Because the seeds include well-known competitive-programming problems, some generated variants may overlap conceptually with public evaluation sets, risking train-test leakage and inflating performance numbers.
4. Data in rStar-Coder comes with an extensive chain-of-thought trace, yet the paper never ablates (i) dataset size and (ii) presence/absence of CoT supervision separately. As a result, it is hard to know whether the reported gains are driven mainly by the additional reasoning signals or simply by more data. If CoT is the dominant factor, similar improvements might be achievable with a far smaller corpus, weakening the case for synthesising such a large number of competitive-programming problems.
5. The validate_test_input function may also be incorrect, and there is no process in place to ensure its correctness.

---

> ### Author Rebuttal · Authors · 2025-07-31
>
> Thank you for your thoughtful and constructive feedback. We sincerely appreciate your recognition of our contributions. Below, we address each of your question.
>
> > Beyond the manually curated seed problems, your pipeline still relies on transforming existing tasks to create new ones. Could you clarify why this dependence is necessary? Is the method infeasible for problems that lack any reference solution, and does this imply that fully unconstrained scaling-up of code datasets is out of reach?
>
> **Response**: Thank you for your insightful question. We address each part below:
>
> 1. Why rely on seed problems?
>
> Unlike general coding tasks such as HumanEval/SWEBench, where new prompts can be bootstrapped from real-world codebases, competitive programming problems are uniquely challenging. They require **significant algorithmic reasoning** and **creative thinking** capabilities, which remains challenging even for top reasoning models.
>  For example, OpenAI's latest reasoning model placed second in a recent AtCoder contest (a key data source included in our dataset) [1], while human coders still dominate. As such, generating valid, solvable, and non-trivial problems entirely from scratch remains beyond current LLM capabilities.  Seed problems serve as essential scaffolds in this setting.
>
> [1] 7/27, 2025.  Human Coder Edges Out OpenAI in Atcoder Finals, but AI Gains Ground. https://shiawaves.com/english/news/science/ai-news/129745-human-coder-edges-out-openai-in-atcoder-finals-but-ai-gains-ground/
>
> 2. Is the method infeasible for problems lacking reference solutions?
>
> At present, our pipeline requires both the seed problem and the step-by-step solution. The reason is still due the high-difficulty of these competition-level problems. Without solutions to expose the underlying knowledge and reasoning skills being tested, general-purpose models (e.g., GPT-4o) lack the grounding needed to create new problems that preserve both difficulty and solvability.
>
> 3. Does this imply that fully unconstrained scaling-up is out of reach?
>
> We view this dependence as a current model limitation rather than a fundamental barrier. As reasoning models continue to improve,  future pipelines could bootstrap new problems without explicit references by verifying candidate solutions.
>
>
> We hope this explanation addresses your questions and provides a clearer understanding of our approach. Thank you again for your valuable feedback, and we welcome any additional questions or suggestions you might have!
>
>
> >rStar-Coder’s problems are overwhelmingly competitive-programming style (algorithmic puzzles with full I/O control). It is unclear how well the pipeline generalises to more heterogeneous software-engineering tasks—e.g. SWE-bench bug-fixing, class-level code synthesis, or front-end/web-UI challenges—where test-case design and solution formats differ substantially.
>
> **Response**: Thank you for the thoughtful question. While our current work targets competitive programming,  core components like input generation, mutual verification are domain-agnostic and can be adapted to broader code-generation tasks.
>
> That said, competitive programming tasks differ significantly from general software engineering benchmarks. In SWE-bench-style tasks, prompts and test cases often come from natural language description or code bases. In contrast, competitive programming problems are self-contained, require **end-to-end algorithmic reasoning**, and demand **strict I/O control** with a wide range of test case difficulties.  These differences make direct generalization non-trivial and motivate our domain-specific design.
>
> We do not claim a universal solution. Instead, our goal is to address the underexplored challenge of synthesizing
>  high-quality, competition-level problems to advance LLM code reasoning capabilities, where the training data is highly scarce. To our knowledge, rStar-Coder is **the first large-scale pipeline capable of synthesizing such high-difficulty code problems with quality guarantees and diverse test cases**.
>
> We agree that extending to more diverse domains is an exciting future direction, and leave this as our future works.
>
>
> >Did you perform deduplication between the training corpus and each evaluation set to prevent train–test leakage? If so, please describe the procedure and its impact on the reported results.
>
> **Response**: Yes, as described in Section 3.3,  we carefully performed deduplication between our training samples and all evaluation sets to minimize train-test leakage.
>
> Specifically, we follow the widely-adopted n-gram matching approach as in prior works [1][2], removing any training problems that share a 16-gram overlap with each evaluation benchmark, The table below summarizes the  number of removed samples based on the n-gram overlap:
>
> |benchmark|# removed samples|
> | :--:   |:--:   |
> |Livecodebench | 216|
> |Humaneval+| 45|
> |MBPP+| 4|
> |USACO 2025| 47|
> |total| 312|
>
> We will include these numbers in the revision for clarity.
>
> [1] Qwen2.5-Math technical report: toward mathematical expert model via self-improvement
>
> [2] s1: Simple test-time scaling
>
> >Data in rStar-Coder comes with an extensive chain-of-thought trace, yet the paper never ablates (i) dataset size and (ii) presence/absence of CoT supervision separately. As a result, it is hard to know whether the reported gains are driven mainly by the additional reasoning signals or simply by more data. If CoT is the dominant factor, similar improvements might be achievable with a far smaller corpus, weakening the case for synthesising such a large number of competitive-programming problems.
>
> **Response**: Thank you for the insightful question! Final code reasoning performance depends on multiple factors: overall data quality, dataset size, and the quality of CoT traces. While we didn’t include full ablations in the original paper, we now provide new results to ablate these factors.
>
> 1) **Data quality > Size**: Despite having fewer examples (580K), rStar-Coder outperforms models trained on larger datasets (736K in OCR, 1M in OpenThinker2), as shown below. This highlights the importance of overall data quality (i.e., problem difficulty, diversity and reasoning quality) over raw scale.
>
>
> |model|dataset size|LiveCodeBench | HumanEval|HumanEval+|MBPP|MBPP+|
> | :--:   |:--:   |:--:   |:--:   |:--:   |:--:   |:--:   |
> |rStar-Coder-7B| **580K**| **57.3**| **95.9**| **90.8**| **87.9**| **74.0**|
> |OCR-Qwen-7B-Instruct| 736K| 51.3| - | - | -|-|
> |OpenThinker2-7B| 1M | 37.4| 92.7| 87.7| 86.9| 73.9|
>
> These results suggest that data quality, problem difficulty, and reasoning supervision play a much more significant role than dataset size alone.
>
> 2) **Dataset size ablation**: We incrementally added synthetic problems to a fixed seed dataset.  As shown in the table below, reasoning performance on LiveCodeBench improves consistently as dataset size increases:
>
> |dataset| LiveCodeBench| HumanEval|HumanEval+|MBPP|MBPP+|
> | :--:   |:--:   |:--:   |:--:   |:--:   |:--:   |
> |Seed only | 46.8| 89.6| 83.6| 86.5| 72.7|
> |Seed +50K synthetic |54.0 |94.4 |88.3|**91.2**|**76.5**|
> |Seed +100K synthetic  |55.1 |94.2 |87.9|90.6|75.4|
> |Seed +200K synthetic |56.2 |93.8 |88.0|89.2|74.5|
> |Seed +all synthetic (rStar-Coder dataset) |**57.3**|**95.9** |**90.8**|87.9|74.0|
>
> 3) **CoT supervision ablation**: We removed the CoT traces from the seed-only set, retaining only the final Python solutions.  As expected, performance dropped sharply:
>
> |dataset| LiveCodeBench|  HumanEval|HumanEval+|MBPP|MBPP+|
> | :--:   |:--:   |:--:   |:--:   |:--:   |:--:   |
> |Seed only (long cot traces) | 46.8| 89.6| 83.6| 86.5| 72.7|
> |Seed only (no cot traces, python code solution only)| 13.6| 70.1| 64.6| 82.3| 68.3|
>
> These findings reinforce our design: dataset scale and CoT are both important, but neither alone is sufficient. Strong performance stems from the combination of high-quality CoT reasoning traces, diverse problem coverage and difficulties, and sufficient scale.
>
> We will include these results in the revision. Thank you again for your suggestion.
>
>
> >The validate_test_input function may also be incorrect, and there is no process in place to ensure its correctness.
>
> **Response**: Thank you for your insightful comment. We agree that the correctness of the validate_test_input function is crucial. In practice, two types of issues may arise if this function is incorrect:
>
> 1. Structurally invalid inputs (e.g., wrong types of malformed structures) are automatically filtered during execution, as they cause runtime errors.
>
> 2. Inputs that violdate problem-specific constraints are harder to detect and not fully handled. However, we observe the current implementation to be highly reliable in practice.  As shown in Table 5 (Left) (original paper), across 3150 generated test cases for seed problems, **96.8%** test input-output pairs are accepted by their reference solutions, indicating that the validate_test_input function is effective in practice.  We’ll clarify this and explore stronger input validation mechanisms as part of future work.
>
> >The authors did not include a Limitations section in the paper. My suggestion is to refer to the Questions part above.
>
> **Response**: Thank you. We would like to clarify that we **do include a limitation section** in the appendix (Appendix A, page 13). To improve visibility, we'll move it into the main text. Additionally, based on your feedback, we will expand the discussion to include the following limitations:
>
> 1) Our current pipeline is specialized for competitive programming problems. Adapting it to more open-ended software engineering tasks (e.g., bug fixing, pull request generation) would likely require special modifications.
>
> 2) Our validate_test_input function may not fully capture all problem-specific constraints, though we see strong empirical results (96.8% pass rate) as shown in Table 5.
>
> We sincerely appreciate your suggestions and are happy to provide further clarification if needed.

---

> > ### Comment · Reviewer_gnwi · 2025-08-05
> >
> > Thank you for your detailed response. Unfortunately, parts (2) and (3) of Question 1 were not fully addressed. While I agree that the model may struggle to generate valid question–answer pairs, one can usually sample many candidate solutions and design metrics to reject invalid data.
> >
> > Your explanation for Question 2 is acceptable, and my remaining three questions have been satisfactorily resolved. I therefore keep my score unchanged. In particular, the idea of using multiple candidate solutions to verify test correctness has already been explored in several prior works, which lessens the incremental contribution of this submission.

---

> ### Author Response · Authors · 2025-08-06
> **Response to Reviewer gnwi**
>
> Thank you again for your thoughtful comments. We appreciate the time you have taken to review our work and provide valuable feedback! We're happy that we have addressed most of your concerns. Below, we would like to clarify a few points regarding your remaining concerns.
>
> **Response 1/2**: We would like to clarify our intent regarding parts (2) and (3) of Question 1, as there may have been a misunderstanding. In our data synthesis pipeline, we **do not ask the model to generate question–answer pairs**. Instead, we start from a seed problem and its reference solution, and prompt the model to generate new problems based on this reference. The reference solution helps the LLM understand the key concepts and reasoning skills involved, ensuring that the generated problems remain solvable and of appropriate complexity.
>
> Without the reference solution, we found that most generated problems were either too simple or unsolvable. Although many candidates can be produced, most are filtered out during mutual verification, leading to limited usable data and significant waste of GPT resources during sampling and filtering.
>
> For example, starting from Codeforces Problem 1332G (No Monotone Triples), removing the reference caused the model to randomly modify the problem statement. This often led to trivial or unsolvable tasks. Below is one such example generated by GPT-4o based solely on the original problem, which turns out to be NP-complete and unsolvable.
>
> >Alice is given a sequence of integers a of length (n). She defines a triple ((i, j, k)) as a balanced triple if:
>
> > 1.	(1 \le i < j < k \le n);
>
> > 2.	(a_i + a_k = 2 \cdot a_j), i.e., (a_j) is the average of (a_i) and (a_k).
>
> > Alice is tasked with finding subsequences b of (a_L, a_{L+1}, \dots, a_R) for each query ((L, R)) such that (|b| \geq 3) and b contains no balanced triples.
>
> > Among all valid subsequences, Alice wants the one with the maximum possible length. If no such subsequence exists, output 0
>
> > Input Format:
>
> > The first line contains two integers (n, q):
> (3 \leq n \leq 2 \cdot 10^5), the length of the sequence and the number of queries.
>
> >	The second line contains (n) integers:
> (a_1, a_2, \dots, a_n) ((1 \leq a_i \leq 10^9)), the sequence elements.
>
> > Each of the next (q) lines contains two integers (L, R):
> (1 \leq L < R \leq n), the range of the query, with (R - L \geq 2).
> ________________________________________
>
> >Output Format:
>
> >For each query:
>
> >	If no subsequence satisfies the conditions, output a single 0.
>
> >	Otherwise, output an integer (k) ((k > 2)) — the length of the subsequence, followed by (k) indices (i_1, i_2, \dots, i_k) such that (L \leq i_1 < i_2 < \dots < i_k \leq R).
>
> >If multiple maximum-length answers exist, print any of them.
>
>
> **Response 2/2)**: Regarding your comment on the use of multiple candidate solutions for test verification, we acknowledge that similar ideas have appeared in prior work. However, we would like to clarify that this is not the main technical contribution of our submission. It is used as a reliable and practical component within a broader pipeline.
>
> Beyond mutual verification for labeling test outputs and verifying solutions, a more important aspect of our work is a three-step pipeline that reliably generates valid test inputs that cover a range of difficulty levels. To our knowledge, this capability has not been demonstrated in prior work. This design enables more fine-grained evaluation and more effective data construction for code reasoning tasks.
>
> We also believe the problem we address, competitive code problem synthesis to advance LLM code reasoning, is both important and underexplored in the open community. Our dataset provides a valuable resource for this direction.
>
> We will clarify these points further in the final version. Thank you again for your feedback!

---

### Official Review · Reviewer_6WNE · 2025-07-03

**Clarity:** 3
**Significance:** 3
**Originality:** 2
**Rating:** 5
**Confidence:** 4

**Summary:**

The paper introduces a large scale pipeline to synthesize large scale difficult problem and solution pairs. It includes proper seeding with selection, carefully prompted model for problem synthesis, test generation and solution generation. Many steps include quality based filtering with careful design. The authors produced 580k examples, trained on which models of different sizes attain very good performance in the weight size class.

**Questions:**

- The final datasets is very large. Would the author be able to provide some taxonomy or other global description of the data so it's easier to understand what is going on.
- [nit] Dual verification in Table 5 is "mutual verification" i believe?

**Ethical Concerns:**

["NO or VERY MINOR ethics concerns only"]

**Final Justification:**

As discussed, I am keeping my original score accept, as I find the paper despite not the most novel, quite useful and solid.

**Limitations:**

yes

**Quality:**

3

**Strengths And Weaknesses:**

The paper is well-written, easy to follow and intuitive. A full pipeline for generating large set of code reasoning problems are carefully designed and well designed. The final synthetic dataset has been shown to substantially improve coding capabilities of LLMs. The ablation results are also interesting in understanding the utility of the synthesis pipeline stages.

While it is obvious that synthesizing coding problems with seeds and solutions with strong proprietary models, e.g. magicoder, for learning is a relatively well-known approach, the specific design in the paper and the strong performance generally outweigh the novelty issue.

The use of GPT-4o may limit the generality of the method as any other methods that uses frontier models.

---

> ### Author Rebuttal · Authors · 2025-07-31
>
> Thank you for your thoughtful and positive feedback on our work! We sincerely appreciate your insights and your recognition of our contributions. Below, we address your specific comments.
>
> > Would the author be able to provide some taxonomy or other global description of the data so it's easier to understand what is going on.
>
> **Response**: Thank you for the helpful suggestion.  In the final version, we will include a detailed taxonomy in the appendix to provide a clearer global view of our dataset. Below is a preview,  organized along three key dimensions: algorithmic task types, file size breakdown, and solution token length distribution.
>
> |Algorithmic Task Types|
> | :--:   |
> | 2-sat, binary search, bitmasks, brute force, chinese remainder theorem, combinatorics, constructive algorithms, data structures, dfs and similar, divide and conquer, dp, dsu, expression parsing, fft, flows, games, geometry, graph matchings, graphs, greedy, hashing, implementation, interactive, math, matrices, meet-in-the-middle, number theory, probabilities, schedules, shortest paths, sortings, string suffix structures, strings, ternary search, trees, two pointers |
>
>
> |Total file size (compressed)| seed problems&solutions | seed test cases| synthetic problems&solutions|synthetic test cases|
> | :--:   |:--:   |:--:   |:--:   |:--:   |
> |**442.1G** | 12G|167G |6.1G | 257G|
>
> |Token length distribution (Solution Traces)| # of solutions  | percentage|
> | :--:   |:--:   |:--:   |
> |<4k|118593|20.3%|
> |4k-8k|111421|19.1%|
> |8k-16k|233002|39.9%|
> |>16k|120748|20.7%|
>
>
> We hope this taxonomy provides a helpful overview of the dataset's scope and structure. We are committed to releasing the full dataset to support open research and advance progress in code reasoning.
>
> > Dual verification in Table 5 is "mutual verification" i believe?
>
> **Response**: Thank you for pointing out this. Yes, "dual verification" in Table 5 should be consistently referred as "mutual verification". We will revise this and carefully proofread the final version for consistency.
>
> > The use of GPT-4o may limit the generality of the method as any other methods that uses frontier models.
>
> **Response**: Thank you for your question. While we used GPT-4o in our main experiments, several key steps in the pipeline (such as new problem synthesis) are model-agnostic. We have verified that smaller open-source models like Qwen2.5-72B-Instruct can also generate high-quality, solvable problems using our synthesis method. This indicates that our approach generalizes beyond GPT-4o and can be reproduced or extended using publicly available alternatives.  We will include this discussion in the revision for further clarity.
>
>
> Below is an illustrative example comparing a seed problem and a new synthetic problem generated by Qwen2.5-72B-instruct:
>
> >**Original seed problem**:
>
> >We have a grid with H rows and W columns of squares. Snuke is painting these squares in colors 1, 2, ..., N. Here, the following conditions should be satisfied:
>
> >* For each i (1 ≤ i ≤ N), there are exactly a_i squares painted in Color i. Here, a_1 + a_2 + ... + a_N = H W.
> >* For each i (1 ≤ i ≤ N), the squares painted in Color i are 4-connected. That is, every square painted in Color i can be reached from every square painted in Color i by repeatedly traveling to a horizontally or vertically adjacent square painted in Color i.
>
> >Find a way to paint the squares so that the conditions are satisfied. It can be shown that a solution always exists.
>
> >Constraints
>
> >* 1 ≤ H, W ≤ 100
>
> >* 1 ≤ N ≤ H W
>
> >* a_i ≥ 1
>
> >* a_1 + a_2 + ... + a_N = H W
>
> >Input
>
> >Input is given from Standard Input in the following format:
>
> >H W
>
> >N
>
> >a_1 a_2 ... a_N
>
> >Output
>
> >Print one way to paint the squares that satisfies the conditions. Output in the following format:
>
> >c_{1 1} ... c_{1 W}
>
> >:
>
> >c_{H 1} ... c_{H W}
>
> >Here, c_{i j} is the color of the square at the i-th row from the top and j-th column from the left.
>
> >Examples
>
> >Input
>
> >2 2
>
> >3
>
> >2 1 1
>
>
> >Output
>
> >1 1
>
> >2 3
>
> >**New synthetic problem by Qwen2.5-72B-Instruct**:
>
> >You are given a rectangular garden with R rows and C columns. Each cell in the garden can be planted with one of K types of flowers. The goal is to plant the garden such that:
>
> >	Each type of flower is planted exactly b_1, b_2, ..., b_K times, where the sum of b_1, b_2, ..., b_K equals R * C.
>
> >	Each type of flower forms a contiguous block, meaning that all cells with the same flower type are connected either horizontally or vertically.
>
> >Your task is to provide a planting scheme for the garden that meets the above conditions. It is guaranteed that a valid solution exists.
>
> >Constraints:
>
> >	1 ≤ R, C ≤ 100
>
> >	1 ≤ K ≤ R * C
>
> >	1 ≤ b_i ≤ R * C
>
> >	b_1 + b_2 + ... + b_K = R * C
>
> >Input: The input consists of three lines:
>
> >	The first line contains two integers R and C, the number of rows and columns in the garden.
>
> >	The second line contains an integer K, the number of flower types.
>
> >	The third line contains K integers b_1, b_2, ..., b_K, the number of times each flower type should be planted.
>
> >Output: Output R lines, each containing C integers representing the flower type planted in each cell. The flower types are numbered from 1 to K.
>
>
> >Input:
>
> >3 5
>
> >5
>
> >1 2 3 4 5
>
> >Output:
>
> >1 4 4 4 3
>
> >2 5 4 5 3
>
> >2 5 5 5 3

---

### Comment · Area_Chair_UFoF · 2025-08-04
**Gentle Reminder: Please Reply to Authors’ Responses (Only if Not Yet Done)**

Dear Reviewers,

As the discussion deadline approaches, may we kindly ask you to review the authors’ responses and post a constructive reply—unless you have already done so, in which case please kindly disregard this gentle reminder.

Your thoughtful engagement is deeply appreciated and essential to a fair and timely process. With sincere thanks for your continued dedication.

Area Chair

---

### Decision · Program_Chairs · 2025-09-17

**Decision:**

Accept (poster)

**Comment:**

Summary of the paper: This paper presents rStar-Coder, an end-to-end pipeline that converts 37.7 K expert-written competitive-programming seeds into 418 K high-difficulty code-reasoning problems together with 580 K long chain-of-thought solutions.  It claims that (i) a two-stage, parameter-controlled test-case generator plus a mutual-verification scheme attains 96.8 % labeling accuracy without human intervention; (ii) fine-tuning even small open-weight models (e.g., 7 B-parameter Qwen2.5) on this data yields state-of-the-art or better results on LiveCodeBench, USACO, and general code tasks, sometimes surpassing frontier models; and (iii) ablations show that each component—seed selection, problem synthesis, test-case diversity, and mutual verification—contributes measurably to final performance, demonstrating that data quality can trump model scale.

Strengths of the paper:
1. Large-scale, validated resources: 418 K problems and 580 K verified solutions constitute a new public resource for training and evaluating code-reasoning LLMs.
2. Methodological novelty: a two-stage test-generation toolkit that exposes scale-controlling parameters plus a cross-checking mutual-verification protocol achieves >96 % accuracy, going beyond prior LLM-based “hallucinate-and-pray” approaches. Each pipeline stage is evaluated, giving the community clear guidance on what matters.
3. Strong empirical gains: fine-tuned 7B models jump from 17 % to 57 % pass@1 on LiveCodeBench and solve more USACO Gold problems than some frontier models, providing convincing evidence that data quality drives reasoning.
4. Clarity and reproducibility: the paper is well-written, the pipeline is intuitive, and the dataset will be released (Please do so after acceptance, I found sampled data are attached in the supplementary materials).

Weaknesses of the paper: Following the rebuttal, the authors have convincingly addressed the concerns raised by the three reviewers. Reviewer 6vKP’s remaining criticisms center on presentation—specifically, the use of misleading or inaccurate terminology, an insufficient discussion of limitations, and overlooked related work. The newly added experiments appear to resolve the concern about the unverified majority-vote assumption, and these presentation issues do not, in my view, compromise the paper’s core contribution. I therefore recommend that the authors incorporate the experimental results and clarifications from the rebuttal into the camera-ready version.

Reasons for the decision: This paper delivers both a significant resource (a 418 K-problem dataset with validated solutions) and a technically solid, well-evaluated pipeline for creating it.  The empirical improvements are significant and seem to be reproducible, and the ablations convincingly attribute these gains to design choices rather than sheer scale or proprietary tricks. Three reviewers appreciate the novelty, effectiveness, and comprehensive experiments of our approach, acknowledging its well-founded methodology and strong performance. Given the strong reviews from them and the clear community value, I think this paper meets the bar for acceptance.